# A pH-gated conformational switch regulates the phosphatase activity of bifunctional HisKA-family histidine kinases

Yixiang Liu[1], Joshua Rose[2], Shaojia Huang[3,4], Yangbo Hu[3], Qiong Wu[1], Dan Wang[1,4], Conggang Li ⓘ [1], Maili Liu[1], Pei Zhou ⓘ [2] & Ling Jiang[1]

Histidine kinases are key regulators in the bacterial two-component systems that mediate the cellular response to environmental changes. The vast majority of the sensor histidine kinases belong to the bifunctional HisKA family, displaying both kinase and phosphatase activities toward their substrates. The molecular mechanisms regulating the opposing activities of these enzymes are not well understood. Through a combined NMR and crystallographic study on the histidine kinase HK853 and its response regulator RR468 from *Thermotoga maritima*, here we report a pH-mediated conformational switch of HK853 that shuts off its phosphatase activity under acidic conditions. Such a pH-sensing mechanism is further demonstrated in the EnvZ-OmpR two-component system from *Salmonella enterica* in vitro and in vivo, which directly contributes to the bacterial infectivity. Our finding reveals a broadly conserved mechanism that regulates the phosphatase activity of the largest family of bifunctional histidine kinases in response to the change of environmental pH.

[1] Key Laboratory of Magnetic Resonance in Biological Systems, State Key Laboratory of Magnetic Resonance and Atomic and Molecular Physics, National Center for Magnetic Resonance in Wuhan, Wuhan Institute of Physics and Mathematics, Chinese Academy of Sciences, Wuhan 430071, China. [2] Department of Biochemistry, Duke University Medical Center, Durham, NC 27710, USA. [3] CAS Key Laboratory of Special Pathogens and Biosafety, Center for Emerging Infectious Diseases, Wuhan Institute of Virology, Chinese Academy of Sciences, Wuhan 430071 China. [4] University of Chinese Academy of Sciences, Beijing 100049 China. Yixiang Liu and Joshua Rose contributed equally to this work. Correspondence and requests for materials should be addressed to L.J. (email: lingjiang@wipm.ac.cn)

Signal transduction by the two-component system (TCS) proteins is the primary mechanism for bacteria to detect environmental changes and orchestrate cellular responses[1–3]. Such a process also plays a critical role in bacterial pathogenesis, for example, by mediating pathogenicity genes during the host infection of *Salmonella* in response to the acidification of the phagocytic vacuole[4].

A typical TCS contains a histidine kinase (HK) and a response regulator (RR). HKs are homodimers consisting of an extracellular sensor domain, a transmembrane connector, and a cytoplasmic portion (HK[cp]) that contains the catalytic activity (Fig. 1a). The cytoplasmic portion of histidine kinases can be further divided into the dimerization and histidine-containing phosphotransfer (DHp) domain and the catalytic and ATP binding (CA) domain[1,5]. About 80% of all sensor kinases belong to the HisKA family[6], including HK853 from *Thermotoga maritima*, EnvZ from *Salmonella*, and others.

After sensing the environmental stimuli, the histidine kinase autophosphorylates a conserved histidine residue in the DHp domain (e.g., H260 of HK853; Fig. 1a) by its catalytic CA domain, which then transfers the phosphoryl group to an aspartate residue of the receiver domain of its cognate response regulator (RR) (e.g., D53 of RR468; Fig. 1a). Such an event triggers the interaction of the phosphorylated RR (phospho-RR) with downstream genes or protein targets for regulation of a variety of cellular functions[1,7].

Due to the essential role of the phosphorylated response regulator in orchestrating the cellular response, its phosphorylation level is tightly controlled. Such a regulation is predominantly achieved through the opposing kinase and phosphatase activities embedded within the module of bifunctional histidine kinase. The conserved histidine residue involved in the phosphoryl transfer process of the DHp domain has also been implicated in the dephosphorylation reaction; the phosphatase activity is additionally affected by conserved Thr and Asn residues of the E/DxxT/N motif (e.g., T264 of HK853; Fig. 1a) immediately adjacent to the conserved histidine residue[8,9]. How a bifunctional histidine kinase switches its kinase and phosphatase activities on and off to maintain a balanced phosphorylation level of the response regulator has remained poorly understood.

Here we report a molecular mechanism that gates the phosphatase activity of the bifunctional histidine kinase HK853, a member of the HisKA family, from *Thermotoga maritima*. Our study reveals a pH-mediated conformational change, involving a sidechain rotameric switch of the catalytic histidine residue that inactivates the phosphatase activity of HK853. Such a mechanism similarly mediates the phosphatase activity of the classical EnvZ bifunctional histidine kinase, also of the HisKA family, from *Salmonella enterica* in vitro; accordingly, the transcriptional response genes of the EnvZ/OmpR TCS are upregulated at low pH in cells, consistent with the pH-gated inactivation of the EnvZ

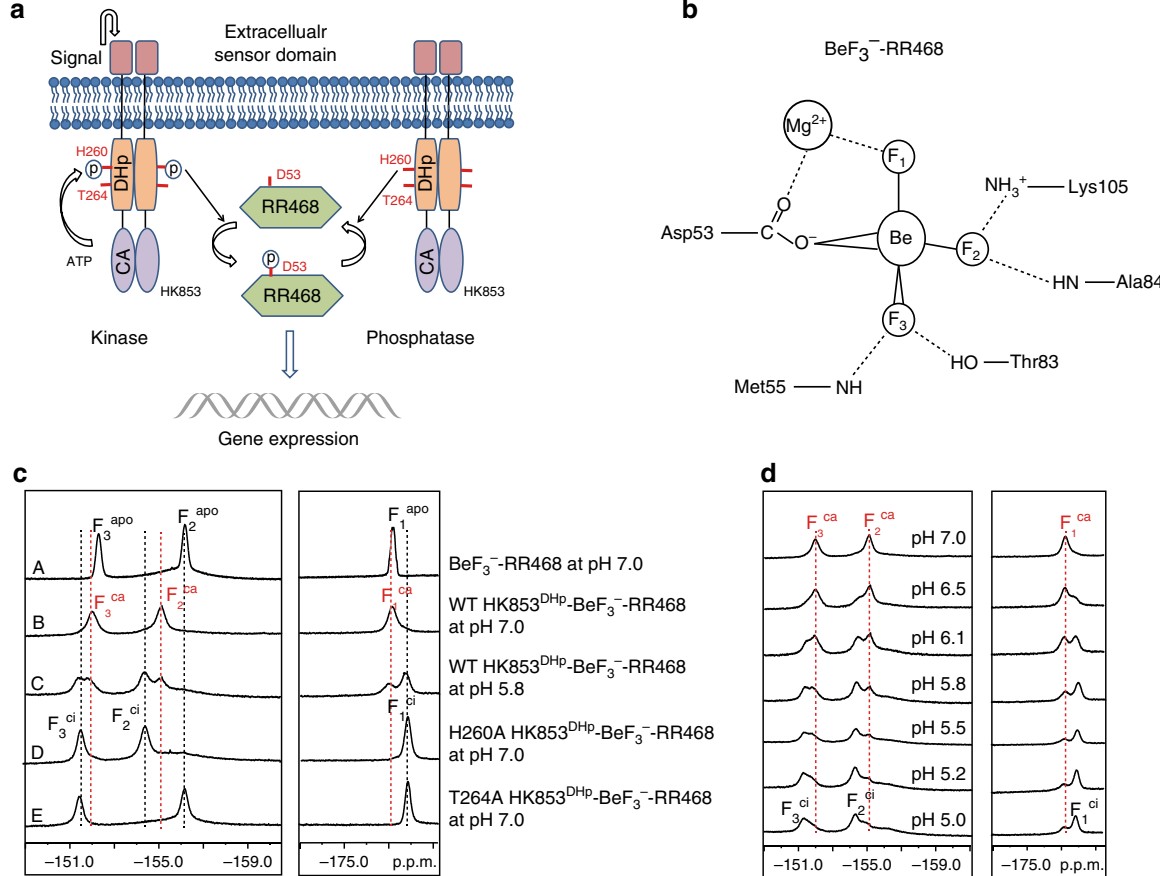

**Fig. 1** $^{19}$F NMR studies reveal multiple conformational states of the HK853-BeF$_3^-$-RR468 protein complex. **a** The signal transduction pathway of the two-component system, exemplified by the bifunctional histidine kinase, HK853, and the downstream response regulator, RR468. **b** A schematic illustration of the coordination of BeF$_3^-$ in the active site of BeF$_3^-$-RR468 (PDB 3GL9). **c** The $^{19}$F NMR spectra of BeF$_3^-$-RR468 without and with HK853$^{DHp}$ and mutants: spectrum A, BeF$_3^-$-RR468 at pH 7.0; spectrum B, the wild-type HK853$^{DHp}$-BeF$_3^-$-RR468 complex at pH 7.0; spectrum C, the wild-type complex at pH 5.8; spectrum D, the H260A HK853$^{DHp}$-BeF$_3^-$-RR468 complex at pH 7.0; spectrum E, the T264A HK853$^{DHp}$-BeF$_3^-$-RR468 complex at pH 7.0. Fluorine signals of *apo* BeF$_3^-$-RR468, the catalytically active HK853$^{DHp}$-BeF$_3^-$-RR468 complex, and the catalytically inactive HK853$^{DHp}$-BeF$_3^-$-RR468 complex are labeled as F$^{apo}$, F$^{ca}$, and F$^{ci}$, respectively. **d** The $^{19}$F NMR spectra of the wild-type HK853$^{DHp}$-BeF$_3^-$-RR468 complex at different pH conditions

phosphatase activity and the subsequent accumulation of the phospho-OmpR. We suggest that the regulatory mechanism revealed by this study may represent a universal pH sensor utilized by the largest family of bifunctional histidine kinases in bacteria and other organisms to detect and respond to the change of the host environment to establish pathogenic infection.

## Results

$^{19}$F NMR reveals two conformational states of the complex. $BeF_3^-$, a well-established phosphoryl analog for studying phosphorylated proteins, has been previously used to generate the phosphorylation mimic of the response regulator[10–12]. Using a similar approach, we generated the $BeF_3^-$-conjugated form of RR468, the cognate response regulator of HK853 from *Thermotoga maritima*, with the beryllium atom forming a covalent bond with the phosphate-accepting residue D53 and with three fluorine atoms (marked as $F_1$, $F_2$, and $F_3$ in Fig. 1b) interacting with the $Mg^{2+}$ ion, sidechains of K105 and T83, and amide groups of M55 and A84 of RR468[5]. When RR468 was mixed with an excess amount of $BeF_3^-$, we were able to detect three broad fluorine signals from the protein-bound $BeF_3^-$ in the active site of RR468 between −180.0 p.p.m. and −150.0 p.p.m. (Fig. 1c, spectrum A), which were distinct from the strong and narrow signals of the unbound beryllium fluoride molecules[13,14].

The identities of the protein-bound fluorine signals were assigned based on the solvent-induced isotopic shift (SIIS) experiment[15–17] (Supplementary Fig. 1): the upper field $^{19}$F signal at −177.18 p.p.m. was the least sensitive to $D_2O$ exchange and was identified as the fluorine atom $F_1$ that coordinates $Mg^{2+}$; the downfield $^{19}$F signal at −152.13 p.p.m. displayed the largest shift when the water ($H_2O$) solvent was replaced with 100% $D_2O$ (Supplementary Table 1 and Supplementary Fig. 1) and, based on the large SIIS effect, was assigned to fluorine $F_3$ that forms hydrogen bonds with the side chain of T83 and the amide group of M55; the central signal at −156.01 p.p.m. experienced an intermediate SIIS effect and was assigned to fluorine $F_2$ that forms a salt bridge with the sidechain of K105 and a hydrogen bond with the amide group of A84. The identity of this fluorine atom was further validated by the lysine sidechain specific H3(N)F experiment that correlated the $F_2$ fluorine signal with the terminal $NH_3$ proton of K105 (Supplementary Fig. 2).

After identifying individual fluorine signals in $BeF_3^-$-RR468, we examined the fluorine signals of $BeF_3^-$-RR468 in complex with the DHp domain of HK853 ($HK853^{DHp}$) that possesses both the kinase and phosphatase activities. The DHp domain of HK853 forms a homodimer and interacts with RR468 at an equal molar ratio[5]. During the titration of $HK853^{DHp}$ at neutral pH, all of the signals in the $^{19}$F spectra became broadened, and a set of new peaks emerged downfield of the original ones, indicating the formation of the $HK853^{DHp}$-$BeF_3^-$-RR468 complex (Fig. 1c, spectrum B). Among the three fluorine signals, $F_2$ was the most perturbed signal with a chemical shift difference of 1.2 p.p.m., indicating that the interaction of the $HK853^{DHp}$ domain significantly alters the salt bridge between $F_2$ and the sidechain of K105.

Interestingly, the $^{19}$F NMR spectrum of the $HK853^{DHp}$-$BeF_3^-$-RR468 complex was highly sensitive to pH changes, a phenomenon that was not observed in $BeF_3^-$-RR468 alone. When pH decreased from 7.0 to 5.0, an additional set of three fluorine signals emerged in the $^{19}$F NMR spectra (Fig. 1c, spectrum C and Fig. 1d). These new signals were located next to the original ones and showed higher intensities when the pH dropped below 6.0, indicating a slow exchange between two distinct conformations of the $HK853^{DHp}$-$BeF_3^-$-RR468 complex. For clarity, we designate the signals of free $BeF_3^-$-RR468 as $F_x^{apo}$

($x = 1, 2, 3$), the signals of the $HK853^{DHp}$-$BeF_3^-$-RR468 complex at pH 7.0 as $F_x^{ca}$ ($x = 1, 2, 3$), and the signals under acidic conditions (pH 5.0) as $F_x^{ci}$ ($x = 1, 2, 3$) (Fig. 1b–d). It is important to note that the circular dichroism spectra of $HK853^{cp}$ and its complex with RR468 remained unchanged between pH 5.0 and 8.0, suggesting that the observed changes of the fluorine signals were not caused by denaturation of the proteins under acidic conditions (Supplementary Fig. 3).

**H260 is involved in the conformational exchange**. A pH-dependent protein conformational change typically occurs in response to the change of the protonation state of a pH-sensitive residue, such as a histidine residue. As the conserved autophosphorylated histidine residue of the bifunctional histidine kinases acts as both the phosphate donor in the kinase reaction and the catalytic residue in the phosphatase reaction[18,19], we generated the corresponding H260A mutation of the DHp domain of HK853 ($HK853^{DHp}$) and verified that this mutation indeed displayed diminished phosphatase activity[5] (Supplementary Fig. 4; uncropped images shown in Supplementary Fig. 5). Surprisingly, the $^{19}$F NMR spectrum of the H260A mutant of the $HK853^{DHp}$-$BeF_3^-$-RR468 complex displayed only three protein-bound fluorine signals that matched the chemical shifts of the $F_x^{ci}$ signals observed in the wild-type protein complex under acidic conditions (Fig. 1c, compare spectra C and D). Furthermore, these fluorine signals no longer displayed pH-dependent changes (Supplementary Fig. 6). Additionally, the signals of the H260A $HK853^{DHp}$-$BeF_3^-$-RR468 complex were much sharper than the signals from the wild-type complex: the half-width of the $F_1$ signal in the wild-type complex was 306 Hz at pH 7.0, whereas the half-width of the same signal in the H260A mutant complex was only 220 Hz; such an observation supports the notion that the pH-dependent conformational exchange mediated by H260 is suppressed or eliminated in this mutant.

**An inactive phosphatase conformation of HK853 at low pH**. As our $^{19}$F NMR studies revealed the presence of a second conformation that is marginally populated at neutral pH, but becomes the predominant state under acidic conditions, we determined the crystal structure of $HK853^{cp}$ and its cognate response regulator RR468 in the presence of the phosphoryl analog $BeF_3^-$ and ATP analog, AMPPNP, under acidic conditions (pH 5.0) in order to visualize this minor conformation at neutral pH. The crystal structure, determined at 2.68 Å resolution (Table 1, a portion of the electron density map is shown in Supplementary Fig. 7), reveals a heterotetrameric complex consisting of two $HK853^{cp}$ and two $BeF_3^-$-RR468. The two anti-parallel helices of the N-terminal DHp domains pack into a central four-helix bundle, with the C-terminal CA domains and the RR468 monomers docking onto and surrounding the central four-helix bundle (Fig. 2a). The AMPPNP is captured in the hydrolyzed form of ADP, as previously observed[5]. The structure of the $HK853^{cp}$-$BeF_3^-$-RR468 complex is close to symmetric, with the backbone rmsds of 0.4 and 0.2 Å between individual monomers of the kinase and response regulator, respectively. The overall architecture of the $HK853^{cp}$-$BeF_3^-$-RR468 complex crystallized at pH 5.0 is similar to the previously reported structure of the $HK853^{cp}$-RR468 complex determined at higher pH (pH 5.6; PDB 3DGE)[5].

However, a close examination of the conserved and catalytically important histidine residue involved in the pH-dependent change of NMR signals reveals distinct conformations. In the previously reported structure of the $HK853^{cp}$-RR468 complex determined at higher pH, the sidechain of $HK853^{cp}$ H260 adopts a *trans* $\chi^1$ rotameric conformation and points toward D53, the

**Table 1 Data collection and refinement statistics for the wild-type and T264A HK853$^{cp}$-BeF$_3^-$-RR468 complex[a]**

| | Wild-type HK853$^{cp}$-BeF$_3^-$-RR468 | T264A HK853$^{cp}$-BeF$_3^-$-RR468 |
|---|---|---|
| *Data collection* | | |
| Space group | C 1 2 1 | C 1 2 1 |
| Cell dimensions | | |
| $a, b, c$ (Å) | 177.97, 98.38, 72.05 | 178.46, 98.34, 71.46 |
| $\alpha, \beta, \gamma$ (°) | 90.0, 110.5, 90.0 | 90.0, 109.8, 90.0 |
| Resolution (Å) | 49.19–2.68 (2.78–2.68) | 45.50–3.63 (3.76–3.63) |
| *R*-meas | 0.081 (0.886) | 0.165 (0.614) |
| *I* /σ*I* | 20.73 (3.14) | 6.56 (2.41) |
| Completeness (%) | 100 (100) | 100 (97) |
| Unique reflections | 32,771 (3254) | 13,230 (1272) |
| Redundancy | 7.7 (7.7) | 4.2 (4.0) |
| *Refinement* | | |
| Resolution (Å) | 49.19–2.68 (2.78–2.68) | 45.50–3.63 (3.76–3.63) |
| Unique reflections | 32,762 (3254) | 13,230 (1272) |
| $R_{work}/R_{free}$ | 0.182/0.222 | 0.208/0.240 |
| No. of atoms | 5714 | 5462 |
| Protein | 5499 | 5398 |
| Ligand/ion | 125 | 64 |
| Water | 90 | 0 |
| Average *B*-factors | 64.55 | 103.22 |
| Protein | 64.45 | 103.16 |
| Ligand/ion | 77.24 | 108.27 |
| Water | 53.23 | |
| R.m.s. deviations | | |
| Bond lengths (Å) | 0.005 | 0.002 |
| Bond angles (°) | 0.81 | 0.56 |
| Ramachandran | | |
| Favored (%) | 95.6 | 93.8 |
| Outliers (%) | 0.7 | 0.3 |

[a]Values in parentheses are for highest-resolution shell

receiving residue of the phosphate group of the response regulator RR468; the imidazole ring of H260 of HK853$^{cp}$ is located within 3.5 Å of a sulfate molecule that mimics the phosphate moiety of the phosphorylated D53; and the overall configuration represents a catalytically active conformation. In contrast, in the structure of the HK853$^{cp}$-BeF$_3^-$-RR468 complex crystallized at lower pH (pH 5.0), where the conformational state in solution is predominantly the F$^{ci}$ state, the sidechain of H260 adopts a *gauche-* $\chi^1$ rotameric conformation (Supplementary Fig. 7), with the imidazole ring of H260 located at 11.2 Å away from the beryllium atom of the BeF$_3^-$-modified D53 of the response regulator RR468; such a configuration would prevent H260 from engaging in the phosphatase activity of HK853 and hence represents a catalytically inactive state (Fig. 2b). Accompanying the sidechain movement of H260, the inter-domain arrangement of the inactive complex also changed. The N-terminal half of the first helix, which contains H260, was less bent compared to that of the catalytically active structure solved at a higher pH, with the location of the N-terminus of the helix shifting about 2.1 Å. Such a difference in the packing of the DHp domain caused a clockwise rotation of the C-terminal CA domain and the response regulator RR468 around the central helices, resulting in distance shifts between the catalytically inactive and active conformations of as large as ~6.9 and ~2.7 Å for the CA domain and RR468, respectively (Fig. 2c, d). These structural observations provide a molecular explanation for the lack of the phosphatase activity of HK853 under acidic conditions shown in the Phos-tag assay (Fig. 2e, f; uncropped images shown in Supplementary Fig. 8a, b, respectively) and suggest that the

phosphatase activity of HK853 is regulated by a pH-gated conformational switch.

**T264 is required for the H260-mediated phosphatase activity.** Beside H260, T264 is another conserved residue that is critical for the transmitter phosphatase activity of HK853. When we titrated the T264A mutant of HK853$^{DHp}$ into the BeF$_3^-$-RR468 solution, the fluorine signal (F$_2$) interacting with K105 of RR468 was unchanged; however, the other two fluorine signals interacting with T83 and Mg$^{2+}$ of RR468 were perturbed and were in good agreement with the fluorine signals of the inactive F$^{ci}$ state of the HK853-RR468 complex (Fig. 1c, spectrum E). These observations suggest that the T264A HK853 mutant still formed a complex with the response regulator RR468, but the critical chemical entity participating in the RR468 K105-fluorine (F$_2$) interaction was absent in the T264A mutant of the HK853-RR468 complex.

As T264 of HK853$^{cp}$ is at least 5 Å away from either K105 or the K105-interacting fluorine atom (F$_2$) of BeF$_3^-$-modified D53, we suggest that the chemical entity causing the chemical shift perturbation of the K105-interacting fluorine atom (F$_2$) is a T264-mediated water molecule, which in turn forms a hydrogen bond with the F$_2$ fluorine atom; furthermore, this water molecule is likely the catalytic water molecule that additionally interacts with H260 of HK853 in the *trans* $\chi^1$ rotameric configuration of the catalytically active state at high pH (Fig. 3, upper right panel). Supporting our model of dual interactions of T264 and H260 with the catalytic water molecule, a sidechain-flipped configuration of H260 captured under the acidic condition in our crystal structure of the HK853$^{cp}$-BeF$_3^-$-RR468 complex would result in a loss of the critical hydrogen bond between H260 and the catalytic water molecule and the perturbation of the chemical shift of the K105-interacting fluorine F$_2$ to the catalytically inactive state (Figs. 1c, 3, lower right panel). Further elimination of the hydroxyl group in the T264A HK853 mutant would result in a complete loss of the catalytic water molecule (Fig. 3, lower left panel), returning the chemical environment of the F$_2$ fluorine atom in the T264A HK853-BeF$_3^-$-RR468 complex to that of the substrate state of BeF$_3^-$-RR468 (Fig. 3, upper left panel). A similar water molecule has been previously hypothesized to interact with a conserved glutamine residue of the DxxQ motif in the nitrogen sensor NarX of the HisKA_3 family based on mutagenesis studies, though the phospho-accepting histidine residue was not thought to be involved in the water interaction[8].

In order to probe the chemical environment of the HK853$^{cp}$ T264A mutant and gain further structural support for our proposed model of dual interactions of HK853$^{cp}$ T264 and H260 with the catalytic water molecule, we crystallized the T264A HK853$^{cp}$-BeF$_3^-$-RR468 complex under the same condition as the wild-type complex at pH 5.0, and determined the structure at 3.6 Å resolution (Table 1, a portion of the electron density map is shown in Supplementary Fig. 9). Consistent with our proposed model (Fig. 3), we found that the structure of the T264A mutant complex indeed captured the same catalytically inactive conformation as the wild-type complex at pH 5.0, with H260 adopting a *gauche-* $\chi^1$ rotameric conformation (Fig. 4a, Supplementary Fig. 9). The overall backbone rmsd between the two structures is 0.32 Å, confirming that the T264A mutation did not affect the binding interface or the stability of the protein complex. However, with the loss of the hydroxyl sidechain in the T264A mutation and with H260 swung away from the active site in a catalytically inactive, *gauche-* $\chi^1$ rotameric conformation (Fig. 4b), none of the water bridging functional groups is retained in the mutated HK853$^{cp}$-BeF$_3^-$-RR468 complex, leaving the F$_2$ fluorine

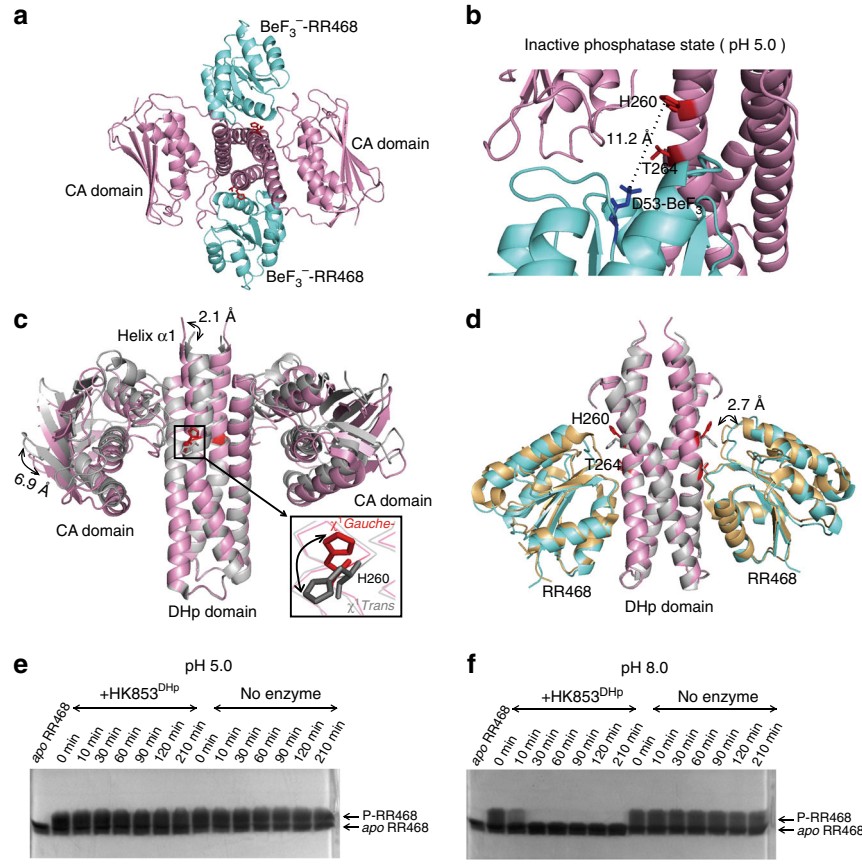

**Fig. 2** Crystal structure of the HK853[cp]-BeF₃⁻-RR468 complex at low pH reveals a catalytically inactive state. **a** The structure of the HK853[cp]-BeF₃⁻-RR468 complex at pH 5.0. The BeF₃⁻-RR468 substrate is shown in cyan, the HK853[cp] dimer in pink, and the sidechains of H260 and T264 of HK853[cp] in red. **b** Structural details of the inactive state of the HK853[cp]-BeF₃⁻-RR468 complex at pH 5.0. The BeF₃⁻-D53 residue of RR468 is shown in blue, and the distance between the ε-N atom of H260 and the Be atom is labeled. **c** A pH-gated conformational switch of HK853[cp]. Superimposed structures of the inactive state (pink, with the sidechain of H260 in red; this structure) and active state (gray, PDB 3DGE) of the HK853[cp]-BeF₃⁻-RR468 complex demonstrate the sidechain flipping of H260 and the movements of the CA domain and the N-terminal α1-helix of the DHp domain. The distances between Cα atoms of C359 of the CA domain and between Cα atoms of R246 of the DHp domain in the two structures are labeled. For visual clarity, RR468 is not displayed. The χ¹ rotameric states of H260 in the two structures are labeled. **d** Structural movement of RR468 in the HK853[cp]-BeF₃⁻-RR468 complex between the inactive state (with RR468 shown in cyan) and active state (with RR468 shown in orange). The distance between Cα atoms of G87 in the two structures is labeled. For visual clarity, the CA domains are not displayed. Structural overlays in **c**, **d** are generated by superimposing the α2-helix and the C-terminal half of the α1-helix (residues of 266–317) of the DHp domain of HK853[cp]. **e**, **f** show the differential phosphatase activities of HK853[DHp] detected by the Phos-tag gel shift assay at pH 5.0 and pH 8.0, respectively

in a chemical environment identical to that in *apo* BeF₃⁻-RR468. Taken together, these observations provide strong support for the dual engagement model of the catalytic water molecule by both H260 and T264 of HK853 (Fig. 3).

**pH-regulated phosphatase activity in the EnvZ-OmpR TCS.** In *Salmonella*, expression of the type III secretion system genes encoded by the *Salmonella* Pathogenicity Island (SPI) 2 is required for systemic infection in mice and is regulated by the *ssrA-ssrB* genes[20] (Fig. 5a). Active transcription of the *ssrA-ssrB* genes occurs after *Salmonella* enters host cells in an acidified vacuole, accompanied by the accumulation of phospho-OmpR, the response regulator of the classical EnvZ/OmpR two-component system[21]. EnvZ belongs to the same HisKA sub-family as HK853 and shares a similar bifunctional motif (Fig. 5b). It has recently been reported that intravacuolar *Salmonella* is acidified in response to external acid stress[4], suggesting that the cytoplasmic pH could be an important regulator of the bifunctional histidine kinase EnvZ that leads to the accumulation of phospho-OmpR.

In order to test this scenario, we first evaluated the pH-dependent phosphatase activity of EnvZ by the Phos-tag assay using the purified cytoplasmic portion of EnvZ (EnvZ[cp]) and the receptor domain of the response regulator OmpR (OmpRr). Similar to HK853, we found that EnvZ[cp] is a less efficient phosphatase under acidic conditions than under neutral pH conditions (Fig.5c; uncropped image shown in Supplementary Fig. 10). Based on these in vitro observations, we went on to examine whether the pH-regulated phosphatase activity of the bifunctional histidine kinase EnvZ is responsible for the accumulation of the phospho-OmpR and transcriptional activation of *ssrA* and *ssrB* in vivo using a quantitative real-time PCR assay. As OmpR is the specific response regulator of EnvZ, the catalytically inactive H243A EnvZ mutant does not generate phospho-OmpR; therefore the background transcriptional levels of *ssrA* and *ssrB* were normalized to one in the *envZ-H243A* mutant strain at pH 5.0 and 7.0, respectively, for comparison with the wild-type strain and the *envZ-T247A* mutant strain under different pH conditions. In a neutral pH environment (pH 7.0), we observed a modest enhancement of the expression of both *ssrA* and *ssrB* due to the presence of

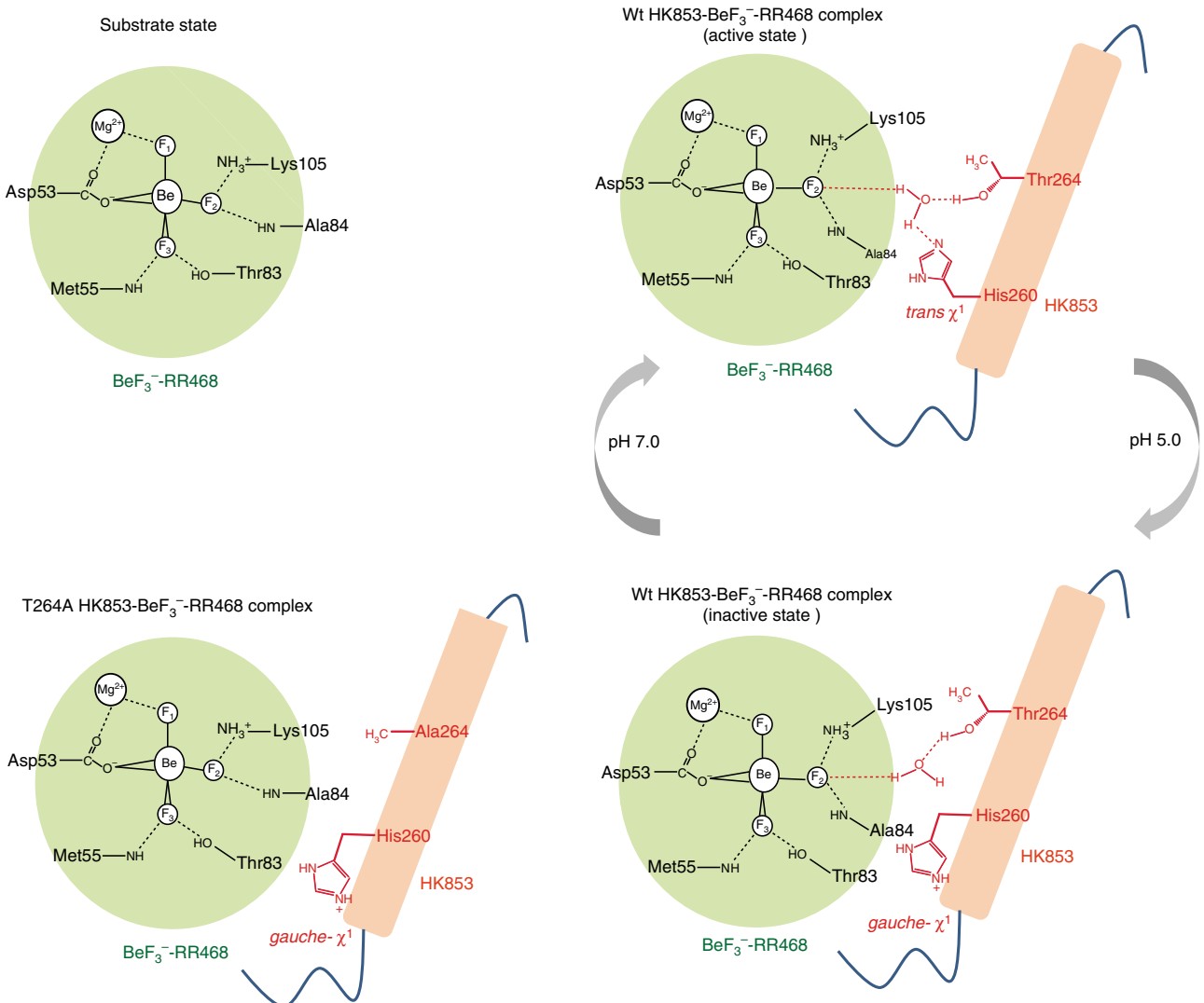

**Fig. 3** Proposed role of the conserved HxxxT motif in the pH-gated phosphatase activity of HisKA. Residues in the active site of RR468 and HK853 are shown in black and red, respectively. In the proposed model, the substrate state (upper left) defines the *apo* state fluorine chemical shifts, which reflects a lack of specific interactions of the $F_2$ fluorine atom with HK853. In the catalytically active state ($F^{ca}$ state, upper right), the $F_2$ fluorine atom (mimicking an oxygen atom of the phosphate group) forms a hydrogen bond with the catalytic water molecule, which is activated by T264 and H260 of HK853, with H260 in the *trans* $\chi^1$ rotameric conformation poised for the catalytic attack of the phosphorus atom of phosphorylated D53 (mimicked by $BeF_3^-$-D53) of RR468. Under acidic conditions, the protonated H260 swings away from the catalytic water molecule, leaving HK853 T264 as the only remaining residue interacting with the catalytic water molecule and the $F_2$ fluorine atom of $BeF_3^-$-RR468, and reflecting a catalytically inactive state of HK853 ($F^{ci}$, lower right). The $F_2$ atom of $BeF_3^-$-D53 in the T264A complex shows a similar chemical environment to the $F_2^{apo}$ state despite the formation of the T264A HK853-$BeF_3^-$-RR468 complex, consistent with the loss of the $F_2$-interacting catalytic water molecule due to the change of H260 $\chi^1$ rotameric conformation to *gauche*- and the elimination of the side chain hydroxyl group from the T264A mutation (lower left)

phospho-OmpR generated by the kinase activity in the wild-type bifunctional EnvZ enzyme (Fig. 5d); however, the enhancements of mRNA levels of *ssrA* and *ssrB* under neutral pH were significantly lower than those under the acidic environment (pH 5.0). Such an in vivo observation is consistent with our in vitro discovery that the phosphatase activity of bifunctional histidine kinases is diminished by a pH-gated conformational switch, resulting in the accumulation of a higher level of phospho-OmpR in the acidic environment. In contrast, although the mRNA levels of *ssrA* and *ssrB* in the *envZ-T247A* mutant strain deficient in the phosphatase activity are enhanced in comparison to those in the wild-type EnvZ strain, the variation in the mRNA levels of *ssrA* and *ssrB* under different pH environments is statistically insignificant, confirming that the observed changes in the mRNA levels of *ssrA* and *ssrB* under the

control of wild-type EnvZ is directly caused by the differential phosphatase activity mediated by a pH-gated conformational switch, but not other factors.

Taken together, our data suggest that a pH-gated inactivation of the phosphatase activity of EnvZ directly regulates the accumulation of the phosphorylated response regulator OmpR and expression of SPI2-encoded genes of the two-component system (*ssrA* and *ssrB*) and the type III secretion system for enhanced bacterial infectivity. Such a model is further supported by the *Salmonella* host invasion experiment: ablation of the kinase activity in the H243A EnvZ mutant eliminated the production of phospho-OmpR and profoundly diminished the macrophage infectivity of the mutant bacterium compared to the wild-type strain, whereas eliminating the phosphatase activity in the T247A EnvZ mutant resulted in an enhanced macrophage

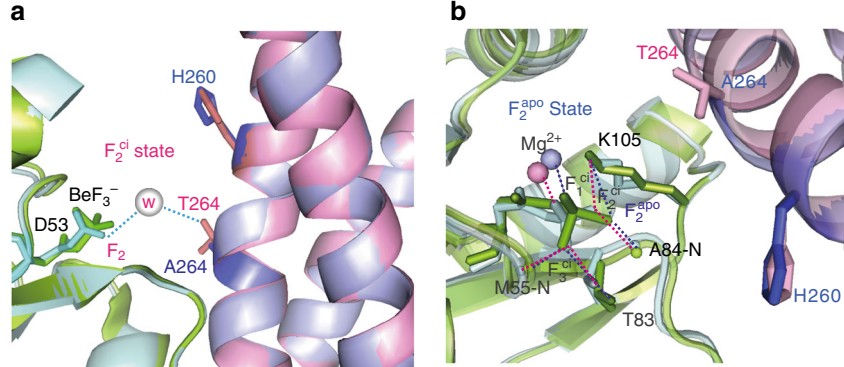

**Fig. 4** Crystal structure of the T264A HK853$^{cp}$-BeF$_3^-$-RR468 complex reveals an "apo" chemical environment for the F$_2$ fluorine of BeF$_3^-$-RR468. **a** Side view of the active site of the T264A HK853$^{cp}$-BeF$_3^-$-RR468 complex overlaid with that of the wild-type complex crystallized at pH 5.0. In the T264A complex, the BeF$_3^-$-RR468 substrate is shown in green and the T264A HK853$^{cp}$ in pale purple, with the sidechains of H260 and A264 of T264A HK853$^{cp}$ in the stick model. In the wild-type complex, the BeF$_3^-$-RR468 substrate is shown in cyan and HK853$^{cp}$ in pink, with the sidechains of H260 and T264 of HK853$^{cp}$ shown in the stick model. The interaction between the F$_2$ atom of BeF$_3^-$-D53 and the hydroxyl group of T264, bridged by the proposed catalytic water molecule (shown in the sphere model) in the wild-type complex at low pH generates the F$_2^{ci}$ state. The loss of the catalytic water and its bridged interactions between the F$_2$ atom of BeF$_3^-$-RR468 and T264 and H260 in the T264A HK853$^{cp}$ mutant generates a chemical environment similar to the F$_2^{apo}$ state. **b** Top view of the active site of the T264A HK853$^{cp}$-BeF$_3^-$-RR468 complex overlaid with that of the wild-type complex crystallized at pH 5.0. Interactions of the fluorine atoms in **a** and **b** are indicated by dashed lines

infectivity due to the accumulation of phospho-OmpR (Fig. 5e, Supplementary Fig. 11).

## Discussion

The bifunctional histidine kinases of bacterial two-component systems possess both phosphoryl transfer activity (kinase) and phosphoryl removal activity (phosphatase) toward their cognate response regulators. Despite the essential roles of these enzymes in regulating bacterial pathogenesis in response to the change of the host environment during infection, the molecular mechanisms that regulate the opposing kinase and phosphatase activities are only starting to be elucidated. In the ChpT-CtrA two-component system of *Brucella abortus*[22], it was suggested that the quaternary structure of the ChpT$_2$–CtrA$_2$ complex enforces an asymmetric mechanism of phosphoryl transfer between ChpT and CtrA, and only one of the two asymmetric binding modes is competent in the phosphotransfer activity of ChpT. However, the molecular mechanism regulating the phosphatase activity in this system remains entirely unclear. Recently, the regulatory mechanism of the bifunctional histidine kinase CckA from *Caulobacter crescentus* was revealed[23]. In this case, binding of the second messenger cyclic di-guanosine monophosphate (c-di-GMP) suppresses the default kinase activity of CckA through noncovalent cross-linking of the catalytic domain with the dimerization histidine phosphotransfer domain. Our study represents a complementary regulatory mechanism of bifunctional histidine kinases that inactivates the phosphatase activity through a pH-gated conformational switch (Fig. 3). It is important to note that the conformation-dependent inactivation process occurs on the slow exchange time scale for NMR measurements (μs-to-ms) and yields two sets of signals corresponding to the conformations of the active and inactivate states between pH 5.2 and 6.5 (Fig. 1d). Such a mechanism is distinct from the general reduction of the catalytic activity due to protonation of the catalytic base, where the fast protonation-deprotonation process yields a single set of population-weighted signals as observed in many enzymes[24]. Interestingly, although the catalytically inactive conformation reported here has not been previously captured in wild-type enzymes, it has been observed in a "rewired" HK853$^{cp*}$-RR468$^*$ complex, in which the interface residues of HK853 and RR468 from *Thermotoga maritima* were

mutated to match those of the two-component system in *Escherichia coli*, PhoR and PhoB, respectively[25]. As these mutations also perturbed the arrangement and relative orientation of the histidine kinase with respect to the response regulator, the critical factor that trapped the chimeric complex into the inactive state has not been isolated. In comparison, by combining solution NMR and crystallographic studies of the wild-type HK853-RR468 complex, we have been able to establish the critical role of pH in regulating the conformation-dependent phosphatase activity of the HisKA family of bifunctional histidine kinases. Whether the inactive conformation of the wild-type HK853-RR468 complex observed at low pH could resemble snapshots of transient intermediates of the phosphotransfer reaction as previously suggested for the rewired complex[25] remains to be investigated.

In summary, our combined solution NMR and crystallographic studies have revealed a pH-gated conformational switch that modulates the phosphatase activity of the largest family of bifunctional histidine kinases. Given the broad conservation of residues involved in this conformation switch, such a regulatory mechanism likely reflects a general pH-response mechanism for the vast majority of the two-component signal transduction systems that function in bacteria, plants and fungi.

## Methods

**Protein preparation and purification**. Plasmids encoding HK853$^{cp}$ (residues 232–489), RR468, EnvZ$^{cp}$ (residues 215–450), OmpR receiver domain (OmpRr; residues 1–125) and His$_6$-MBP-HK853$^{DHp}$ (residues 250–315) were transformed into *E. coli* BL21(DE3) Gold cells (Novagen). For unlabeled protein, the transformed bacterial cells were cultivated in the LB medium containing either ampicillin (100 μg mL$^{-1}$) or kanamycin (50 μg mL$^{-1}$) at 37 °C, and then induced with IPTG (isopropyl β-d-1-thiogalactopyranoside; 1 mM) at 20 °C for 20 h before collecting. $^{15}$N-labeling of RR468 was achieved by growing cells in the M9 minimal medium containing $^{15}$N NH$_4$Cl as the sole nitrogen source.

HK853$^{cp}$ was purified by ammonium sulfate precipitation (40% w/v) followed by anion-exchange and size-exclusion chromatography. RR468 was purified by incubation at 60 °C followed by anion-exchange and size-exclusion chromatography[5,26]. Similar protocols were used to purify EnvZ$^{cp}$ and OmpRr. The H260A and T264A point mutants of HK853$^{DHp}$ were generated by point mutagenesis (primers summarized in Supplementary Table 2). Wild-type and mutant HK853$^{DHp}$ were expressed as His$_6$-MBP fusion proteins. Each fusion protein was purified by using the amylose affinity chromatography and eluted with the buffer containing 20 mM Tris (pH 7.6), 20 mM maltose and 300 mM NaCl. The His$_6$-MBP tag was cleaved by the tobacco etch virus (TEV) protease digestion and removed by passing the cleavage sample through an amylose column in a buffer

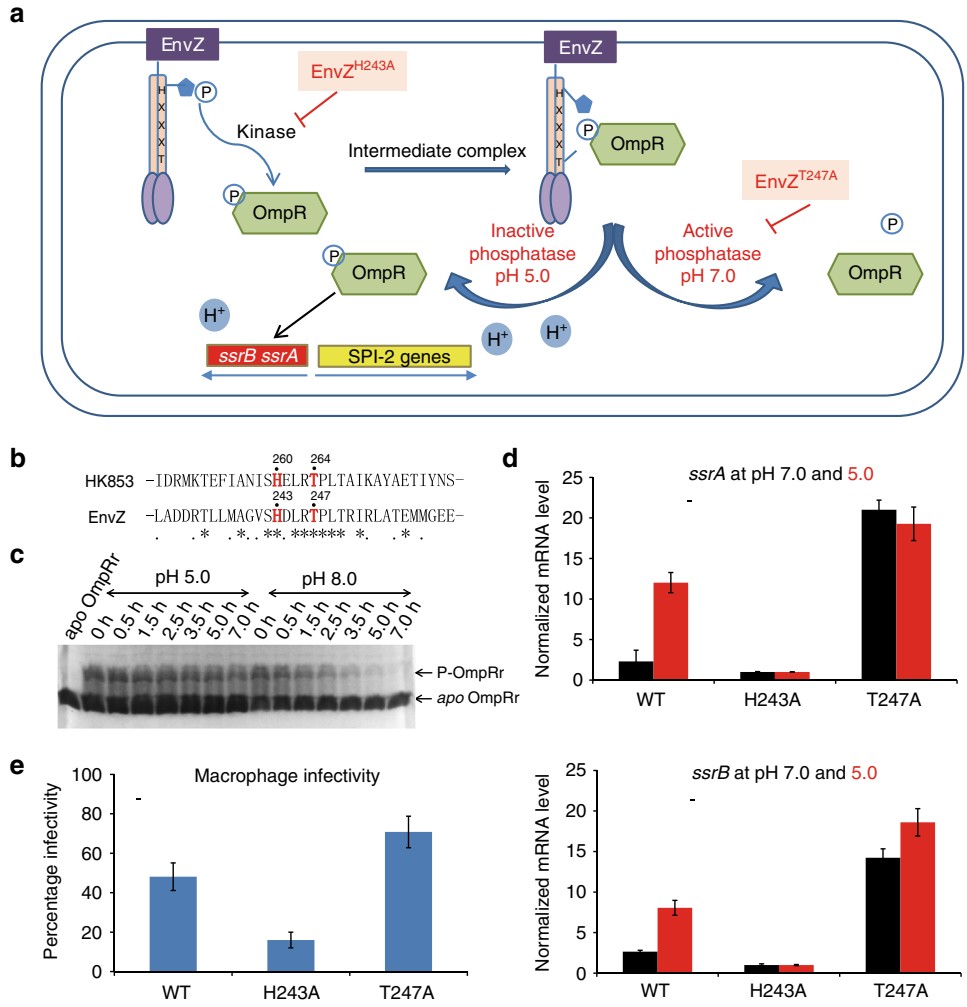

**Fig. 5** pH-gated regulation of the phosphatase activity of the bifunctional histidine kinase EnvZ in *Salmonella*. **a** The schematic illustration depicting the different functions of EnvZ at different pHs during host cell invasion. The H243A and T247A point mutations abolish the kinase and phosphatase activities of EnvZ, respectively. **b** Sequence alignment of the $\alpha_1$ helix in EnvZ and HK853 from the same HisKA subfamily, with the HxxxT motif highlighted in red. Identical and conserved residues are indicated below the sequence alignment. **c** Differential phosphatase activities of EnvZ[cp] detected by the Phos-tag gel shift assay at pH 5.0 and 8.0, respectively. **d** Detection of the expression of *ssrA* and *ssrB* genes by real-time fluorescence quantitative PCR at different pHs. The expression levels at pH 7.0 and 5.0 were shown in black and red, respectively. In each group, the expression level of the H243A mutant is normalized to 1.0 for comparison. **e** Macrophage infectivity of *S. typhimurium* LT2 and mutants. The infectivity was defined as the percentage of infected host cells. Error bars in **d** and **e** represent the standard deviation from quadruplet measurements

containing 20 mM Tris (pH 7.6) and 300 mM NaCl. The target HK853[DHp] bound weakly to the Ni[2+]-NTA column and was separated from the His$_6$-tagged TEV by elution in a buffer containing 50 mM phosphate (pH 8.0), 300 mM NaCl, and 30 mM imidazole. All protein samples were further purified to homogeneity using size-exclusion chromatography.

**NMR experiments**. Protein samples were dissolved in the NMR buffer containing 20 mM HEPES (pH 5.0–7.0), 50 mM KCl, and 10 mM MgCl$_2$, with various levels of D$_2$O. For NMR samples without D$_2$O, a hermetical tube containing D$_2$O was inserted into the NMR tube for the field lock.

BeF$_3^-$-RR468 was generated by adding 5 mM BeCl$_2$ and 50 mM NaF to the RR468 solution, which was incubated at room temperature for 30 min. The protein complexes were generated by adding HK853[DHp] or mutants at a concentration 10-fold higher than the substrate BeF$_3^-$-RR468.

The [19]F NMR experiments[13,14] were performed on a 600 MHz Bruker spectrometer (equipped with a 5 mm CPQCI [1]H/[19]F.-[13]C/[15]N/D Z-GRD cryoprobe) at 298 K, and the chemical shifts were referenced to CFCl$_3$ (the free fluorine signal F$^-$ at −119.0 p.p.m.).

The pulse sequence for the H3(N)F experiment was adapted from the H3(N)P experiment originally designed to analyze the $^{3h}J_{NP}$ coupling[27]. For the [19]F dimension, the carrier position was set to −156.0 p.p.m. The evolution decay in the [15]N dimension was set to 35.0 ms. The chemical shift of lysine sidechain NH$_3$ was assigned by 2D [1]H–[15]N HISQC, 3D (H)CCENH3, and 3D (H)CCNH experiments[28,29].

NMR data were processed with NMRpipe[30] and analyzed by Topspin 3.2.

**Protein crystallization and structural analysis**. The protein complex (10 mg mL$^{-1}$ HK853[cp] and 7.5 mg mL$^{-1}$ RR468) was dissolved in a buffer containing 10 mM Tris (pH 8.0), 7 mM MgCl$_2$, 5 mM BeCl$_2$, 30 mM NaF, and 4 mM AMPPNP[5]. Crystallization of the wild-type complex was carried out by using the sitting drop vapor diffusion method at 20 °C in drops containing equal volumes of the protein complex and mother liquor consisting of 0.1 M citric acid (pH 4.0), 0.8 M ammonium sulfate, with the final pH adjusted to 5.0. For the T264A mutant complex, the mother liquor contained 0.1 M citric acid (pH 5.0), 0.208 M ammonium sulfate, and crystallization was carried out at 15 °C by using the hanging drop vapor diffusion method. The crystals were cryoprotected by a solution containing 50% mother liquor (v/v) and 50% glycerol (v/v) for the wild-type complex and 50% mother liquor (v/v) and 50% sucrose (v/v) for the T264A mutant complex, respectively. The data were collected at the SER-CAT 22-ID and 22-BM lines at Argonne National Laboratory for the wild-type and T264A mutant complexes, respectively; diffraction data were processed with XDS[31] or HKL2000[32] and truncated and scaled with the University of California Los Angeles anisotropy diffraction server[33,34]. The previously reported structure of the HK853[cp]-RR468 complex crystallized at pH 5.6 (PDB: 3DGE)[5] was used as the search model for molecular replacement by using the PHENIX suite[35]. The structural model was iteratively built by COOT[36] and refined by PHENIX. In the crystal structure of the wild-type HK853[cp]-RR468 complex, additional electron density was observed near

one of the two H260 residues, which was interpreted as a glycerol molecule from the cryo-protectant.

**Dephosphorylation assay in vitro.** Samples of RR468 and OmpRr were phosphorylated at a protein concentration of 4 mg mL$^{-1}$ in the presence of 12 mM acetylphosphate at 25 °C for 1 h. The free phosphate was removed by using a GE desalting column. Samples of phosphorylated RR468 and phosphorylated OmpRr were exchanged into desired reaction buffers (20 mM Tris, 10 mM MgCl$_2$, 100 mM NaCl at pH 8.0; and 20 mM NaAc, 10 mM MgCl$_2$, 100 mM NaCl at pH 5.0), flash-frozen, and stored at −80 °C.

A reaction mixture of phospho-RR468 and HK853$^{DHp}$ or mutant enzymes at a molar ratio of 6:1 was incubated at 4 °C for the detection of the phosphatase activity by the native polyacrylamide gel electrophoresis (PAGE) assay, whereas a molar ratio of 13.8:1 was used for the Phos-tag gel assay. The same protocol was used to assay the phosphatase activity of the EnvZ$^{cp}$ and mutants towards phospho-OmpRr. Samples at different time points were taken by flash-freezing at −80 °C. For the native-PAGE detection, gels were run on ice at 90 V for 5 h. The Phos-tag gel was made by adding 75 uM Phos-tag acrylamide and 150 μM MnCl$_2$ into the SDS-PAGE[37]. All the gels used for assays were stained by Coomassie blue.

**Construction of *Salmonella* EnvZ mutant strains.** *S. typhimurium* strains containing the *envZ* H243A or T247A mutations were constructed from the LT2 strain[38,39]. Briefly, the *envZ* gene in LT2 strain was first replaced by a kanamycin resistance gene (*kan*) to obtain the *envZ::kan* strain using the pKD46-mediated gene disruption system[40]. The kanamycin resistance gene in the *envZ:: kan* strain was subsequently replaced with the H243A or T247A mutant of the *envZ* gene using a homologous recombination method as previously described[41]. Detailed procedures are described in Supplementary Methods.

**RNA extraction and qRT-PCR.** Cells of *S. typhimurium* strains were grown in a modified N-minimal medium (MgM) buffered with 20 mM Tris (pH 7.0) or 20 mM MES (pH 5.0) containing 7.5 mM (NH$_4$)$_2$SO$_4$, 5 mM KCl, 0.5 mM K$_2$SO$_4$, 1 mM KH$_2$PO$_4$, 10 mM MgCl$_2$, 2 mM glucose, and 0.1% Casamino acids[4]. When OD$_{600}$ reached 0.5, cells were collected by centrifugation. Total RNA of each collection was extracted with the TRIzol reagent (Invitrogen) according to the manufacturer's protocol. After treatment with RNase-free DNase I (Promega), 2 μg of each RNA sample was used as the template to obtain the cDNA by RNA PCR kit (AMV) ver. 3.0 (TaKaRa). Real-time PCR was performed using Universal SYBR Green Supermix Reagent (Bio-Rad). Results were normalized to the 16S rRNA level. For each strain, wild-type or mutants, all tests were repeated four times independently. Mean values and standard deviations (SD) of these experiments were calculated. Detailed information of PCR primers is listed in Supplementary Information.

**Immunofluorescence detection of macrophage infection.** *S. typhimurium* LT2 cells were grown in the LB medium overnight at 37 °C until OD$_{600}$ reached 1.0. The RAW 264.7 host cells (FineTest, Wuhan) were cultured at 37 °C and then infected by LT2 at a multiplicity of infection of 100 for 30 min followed by PBS washes. The host cells were further incubated for 4 h, then fixed with 4% formaldehyde, permeabilized in 0.2% Triton X-100, and treated with 5% FBS before antibody incubation. LT2 cells were incubated with anti-*Salmonella* Csa-1 primary antibody (KPL) and DyLight 549-conjugated AffiniPure rabbit anti-sheep IgG (EarthOx) second antibody at 4 °C overnight. The host cells were visualized by FITC-conjugated phalloidin (Sigma-Aldrich) that stains actin filaments. The excitation wavelengths of the two different fluorescent dyes were 495 and 552 nm, respectively. The final images were visualized and processed by confocal microscopy.

**Data availability.** The coordinates and structure factors for the X-ray structures of the wild-type HK853$^{cp}$-BeF$_3^-$-RR468 complex and the T264A mutant complex at pH 5.0 have been deposited to the Protein Data Bank (PDB) with accession codes 5UHT and 6AZR, respectively. Other data are available from the corresponding author upon request.

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

## Acknowledgements

We would like to thank Prof. Honggao Yan from the Michigan State University for providing the HK853[DHp] and RR468 plasmids, Mr. Zhongwei Wang from the Wuhan Institute of Virology for assistance with the qRT-PCR experiment, Prof. Lin Guo and Dr. Lizhi Hu from the Wuhan University for help with the infectivity experiment, and Dr. Chenyu Cao from the Northwest A&F University for optimizing the protocols of HK853[DHp] and RR468 purification. Use of the Advanced Photon Source was supported by the U.S. Department of Energy, Office of Science, Office of Basic Energy Sciences, under Contract No. W-31–109-Eng-38. This work was supported by grants from National Key R&D Program of China (#2017YFA0505400 awarded to Dr. Xu Zhang from Wuhan Institute of Physics and Mathematics), Ministry of Science and Technology of China (#2013CB910200 awarded to Dr. Chun Tang from Wuhan Institute of Physics and Mathematics) and the Natural Science Foundation of China (#21573280 awarded to L.J. and #21603268 awarded to Y.L.) and the National Institute of General Medical Sciences (GM115355 awarded to P.Z.).

## Author contributions

L.J., P.Z., and Y.L. designed the project. Y.L. performed the NMR sample preparation, NMR data collection, crystallization experiments of the wild-type HK853[cp]-BeF$_3^-$-RR468 complex and in vitro assays. J.R., Y.L., and P.Z. determined the crystal structure of the wild-type complex. J.R. performed the T264A HK853[cp] mutation, sample preparation, crystallization. J.R. and P.Z. determined the structure of T264A HK853[cp]-BeF$_3^-$-RR468 complex. S.H. and Y.H. performed chromosome mutation and qRT-PCR experiments. Q.W. and Y.L. performed invasion experiments. D.W. helped with sample preparations. Y.L., L.J., P.Z., C.L., and M.L. analyzed the data. Y.L., L.J., and P.Z. wrote the manuscript with critical input from all authors.

## Additional information

**Competing interests:** The authors declare no competing financial interests.

