## [Peer Review File · Nature Communications]

Reviewers' comments:

Reviewer #1 (Remarks to the Author):

The primary hypothesis of the paper by Liu et al is that at pH values lower than 6.0 the dominant population of the HK853-BeF₃-RR468 complex has a different conformation to that at neutral pH and this is important in suppressing phosphatase activity within the complex. A shift in population is evident in the ¹⁹F NMR spectra of the bound fluoroberyllate group. On crystallization of the complex at pH 5.0 a modest change was found in the overall structure of the complex compared with a previous structure determined from a crystal grown at pH 5.6, but a substantial shift in the position of the histidine residue previously implicated in the phosphatase reaction was observed. The phosphatase reaction was shown to be inoperative in the complex at pH 5.0. Overall, I think the authors have presented a very interesting combination of data. My more detailed comments on the manuscript are as follows.

1. I am only loosely acquainted with the TCS field and, as a more general interest reader, I found the introduction provided was rather scant in explanation. For example, I was expecting a compelling argument that a pH switch in phosphatase activity was established to be fundamentally important to the underlying biology. Also, I would suggest including the residue numbers of the relevant residues in this particular complex when introducing the field more generally. It makes a useful reference point for the less familiar reader.

2. The authors do a good job of using NMR data to identify a change in conformation. I think the assignment of ¹⁹F resonances in the RR468 complex is justified given the H₃(N)F data. I'm surprised though when discussing the SIIS values that not all of the H-bond partners are included – what about M55NH and A84NH? The reason to be more inclusive is that changes in these H-bonds will also lead to chemical shift changes between complexes. Also the ¹⁹F NMR spectra should be referenced to CFCI₃, see https://en.wikipedia.org/wiki/Fluorine-19_nuclear_magnetic_resonance.

3. I do not understand to use of the word "gradually" on line 102. Is formation of the complex slow, kinetically? I suspect not given the linewidth argument on lines 132-133, where incidentally the linewidths should be reported as quantitatively as achievable, because the differences are quite hard to see in the figure.

4. The numbering of the fluorines in the pdb file provided and in the manuscript differ. I request that in both the authors label the fluorine bound to Mg as F1, the one H-bonded to K105 and A84 as F2, and the one H-bonded to M55 and T83 as F3. This is a new IUPAC convention that is currently in press and, as yet, unavailable to the authors. However, these changes would make the labeling consistent with the convention for the field in the future.

5. Line 110. To be consistent with Fig 1c "6.8" should read "7.0" since the top spectrum of Fig 1d and spectrum B of Fig 1c look indistinguishable.

6. Line 117. Refer to Fig 1d here also.

7. Line 148. The scenario regarding AMPPNP and ADP needs introducing more fully. Also, why was ADP not simply used for crystallization?

8. Line 149. Should read "close to symmetric"

9. Line 155. This argument is central to the manuscript and needs better support with a figure than

currently. I suggest either an overlay or side-by-side figure with 3dge.pdb highlighting the residues relevant to the model for how phosphoryl transfer is occurring. This will show the movement of H260 better than Fig 2c. I would also like the authors to use terminology other than sidechain flipping for the changes in H260 conformation. This terminology is easily misinterpreted as imidazole ring flipping rather than a larger scale movement of the entire sidechain.

10. Line 182. The comparison of chemical shift changes for the T264A mutant are interesting but the interpretation of their source is not a strong argument. I would suggest either cutting back this section to simply the behavior of the NMR, or leave it to a manuscript supported by a crystal structure of this mutant. It doesn't add greatly to the current manuscript, and the space gained by cutting it would enable a fuller introduction to the study. I would also drop Figure 4 and related text, as the data are not sufficiently strong to elevate this model over others.

11. Line 202. Much of this section should be in the introduction. Indeed, maybe the Salmonella results should come before the Thermotoga results as the former set the scene for the relevance of the latter. I take it the Salmonella system does not crystallize well?

12. On a more general level, any reaction with an imidazole attacking a phosphate is likely to be pH dependent close to and below the pKa of the imidazole. A conformational switch is not needed, even though protonation of the imidazole may induce it. I think the discussion should take this into account unless clear evidence can be presented that the pH conformational shift is the result of protonation away from the active site. Also, the H-bonding between H260 and glycerol in the crystal should be discussed.

Reviewer #2 (Remarks to the Author):

The paper addresses the regulation of the phosphatase activity of bifunctional histidine kinases by pH. Based on NMR spectroscopy results the authors propose that at acidic pH there is a conformational switch that inactivates the phosphatase activity, and they solved the X-ray structure of the complex formed by HK853-RR468 at pH 5.0 where the side chain of the conserved histidine (H260) occupies a position that would preclude its function to promote the hydrolysis of the acyl-phosphate bond in phosphorylated RR469~P. They also prove in vitro that the phosphatase activity of EnvZ is inhibited at acidic pH, and using in vivo experiments they show that the EnvZ/OmpR pathway responds to pH and that it modulates infectivity.

An important body of research performed on different members of the HisKA family indicates that the conserved histidine residue acts as a general base to activate the nucleophilic water molecule (Casino et al, Cell, 2009, Huynh & Stewart, Mol Microbiol, 2011). Therefore it should be expected that the pH (7 vs 5, imidazole pKa 6.02) affects the phosphatase activity. However, the authors never state explicitly if H260 is the protonated residue (only in Figure 4 and its legend the proton is mentioned). To my knowledge, this paper presents the first report on the in vitro influence of the acidic pH on the HK domain, but this can be deduced from the literature, decreasing the novelty required for this journal.

The scope of the discovery is overestimated by the authors, as it is probably applicable only to some groups of histidine kinases. Moreover, both HKs studied in this work (HK853 and EnvZ) are from the same family (HisKA). For example, analysis of different mutants suggests that the phospho-accepting His residue is irrelevant for the phosphatase activity of HisKA_3 family, and given that the authors propose a central role for this histidine residue in the pH regulation, it might be possible that their findings are not reproduced among this entire superfamily.

Besides, a pH sensor residing in a so conserved motif is difficult to justify, since in such case all the HisKAs protein of an organism should be controlled by pH.

The experiments, especially the in vitro ones are very convincing. However a control of structure conservation at pH 5 is strictly needed, to rule out the possibility of enzymatic activity loss due to partial or full denaturation. Circular dichroism experiments should be enough. The in vivo evidence though, is scarce: only one qRT-PCR experiment, with only one mutant.

The finding is highly important to scientist in the field who should consider pH as an important variable to take into account for histidine kinase enzymatic activities, but probably not of broad interest.

In both in vivo experiments, the qRT-PCR and the infection a mutant with which does not bind ATP and therefore with no kinase activity but with intact phosphatase activity should be included.

In the qRT-PCR experiment a better way to show the data would be to normalize the result to the wild-type strain in neutral pH.

An experiment analyzing the in vivo phosphorylated and unphosphorylated forms of OmpR in cultures grown under different pH conditions in wild-type and mutants *Salmonella* strain could increase the in vivo evidence.

Apart from NMR and crystallographic experiments that are usually robust enough and accepted without repetitions, the authors perform qRT-PCR in which they do the experiment twice. They also perform an experiment using confocal microscopy, which they performed twice. The description of the quantification is not in the Materials and Method Section, only in the Figure Legend. No statistical analysis is applied at all in this report. However the differences shown seems to be big enough.

Despite the report is in general well-written the Discussion Section is very poor and should be enriched, putting the new results in context with other report of the HK853-RR468 and EnvZ-OmpR and other HK-RR systems.

An important point that has not been addressed in the manuscript is the influence of pH on the autophosphorylation and phosphotransfer activities of HK853 and EnvZ. If pH mediated a switch, it would be expected to regulate the autophosphorylation and phosphatase activities in opposite directions (i.e. at pH 5.0 HK853 should be an active kinase).

Regarding the crystal structure presented, there are not enough details provided on the interactions that might stabilize the position of H260 at pH 5.0. Also, based on this structure, the authors propose that the configuration adopted by H260 would prevent it from participating in the phosphatase activity, but they do not discuss the possibility that, in fact, protonation itself precludes the catalytic role of H260 and that its side chain flipping is just aleatory or a consequence of relocating the new positive charge introduced in the imidazole ring.

The conclusion of the role of the H260 side chain flipping comes from the comparison of the obtained structure at pH 5.0 with the one from Casino et al, *Cell*, 2009 at pH 5.6. To correlate structure and phosphatase activity the authors therefore should also perform the activity assay at pH 5.6.

Finally, since the authors do not address the possibility of an enhanced kinase/phosphotransferase activity at acidic pH, they do not consider that the conformation crystallized might represent a snapshot of an intermediate state in the phosphotransfer reaction. In a paper published by Podgornaia et. al., *Structure*, 2013 the structure of a complex between rewired HK853 and RR468 is presented,

and the H260 adopts a rotamer that points away from the active site and the authors propose that the complex might represent an earlier state of the phosphatase reaction (when the phosphate is still bound to the RR) or the end of the phosphotransfer reaction (just prior to the complex dissociation). So, in this manuscript the authors should make a reference to the mentioned article and evaluate the similarities/differences between the complexes.

Another comments:

- Page 9, line 183-184. "It has been reported that the T264A mutant of HK853 abolishes the phosphatase activity, though the molecular mechanism remains unclear". Actually, there is evidence that support that the conserved Thr or Asn residue in the E/DxxN/T motif coordinates the nucleophilic water for phosphatase activity (for references see Huynh, Stewart, 2011). The authors then propose the participation of a water molecule, so references to previous observations should be made.
- Page 9, lines 190-192. "... the critical chemical entity participating in the K105RR468-fluorine (F2) interaction was absent in the T264A mutant of the HK853-RR468 complex". Is not it possible to evaluate the presence of the salt bridge by NMR, as it was done with the free RR?
- It should be informed if the initial adhesion of the different Salmonella strains is the same, in order to prove that the results of the macrophage infectivity assay at 4 h is a consequence of different ability to survive within the acidic compartment and not in differences in the initial adhesion and ability to invade the cell.
- The figure legends (the main and the supplementary ones) should be self-explicative. All the important information about the experiment should be summarized here. Please check the Figure Legends Section of the checklist.
- Page 12, lines 250-259. This information is not relevant in this context.
- Figure S3. The phosphatase experiment using the WT HK853 does not show any RR-P at t=0. It would be better if the authors provided a figure where it is shown that the reaction was performed with the RR phosphorylated at the beginning of the assay.
- Page 13, line 263-279. It is not clear either if the modulation of the infectivity by the EnvZ/OmpR pathway is a new finding (which by the way should be mentioned in the Result not the Discussion section) or if it is something already known.

Other minor comments:

- Page 5, line 103. The fact that the phosphatase activity resides in the DHp subdomain should be included as a references (see several report of the review Huynh & Stewart, Mol Microbiol, 2011).
- Page 9, line 179. "... suggest that the bifunctional activity of HK853 is regulated by a pH-gated ...". Strictly, only the phosphatase activity was evaluated, not the kinase.
- Page 10, line 205. "systemic" instead of "systematic".
- Page 11, line 223. "... the catalytically inactive H243A EnvZ mutant does generate phospho-OmpR". This sentence does not make any sense, given that the mutation H243 does not allow the autophosphorylation to occur, and therefore there is no generation of OmpR-P. In fact, the authors also wrote "... ablation of the kinase activity in the H243A EnvZ mutant eliminated the production of phospho-OmpR" (page 13, lines 274-275). Probably the authors wanted to say "does not generate", instead.
- Page 11, lines 235-236. The authors state that the T247 mutant is deficient in the phosphatase activity. They do not demonstrate it in vitro but there are reports of this statement, so they should provide the corresponding references.
- Page 16, line 334. A citation of the report where this crystallization condition was found should be included.
- Figure 3a. EnvZH243A and EnvZT247A element are not clear. As it is drawn it seems inhibition.

Reviewer #3 (Remarks to the Author):

This is an outstanding piece of work, which is described quite succinctly but lucidly. I initially approached the ms. with some skepticism, but found their arguments and data compelling. The authors should be congratulated, and the ms. accepted without further revision.

Reviewer #4 (Remarks to the Author):

This paper deals with bacterial two-component systems (TCS) and addresses the question of how the phosphatase activity of bifunctional histidine kinases of the HisKA family is controlled, using as models the HK853-RR468 TCS from *Thermotoga maritima* in vitro, as well as the *Salmonella* EnvZ-OmpR TCS in vivo. This question is relevant as the mechanisms controlling the balance between the kinase and phosphatase activities of histidine kinases are far to be understood. The main claim of this paper is that a pH-mediated conformational switch inhibits the phosphatase activity of the kinases at low pH. This pH-mediated modulation is independent of the kinase extracellular domain.

Although the work is interesting and well presented, and the data are convincing regarding the pH-mediated phosphatase inhibition and the pH-induced conformational switch, my main concern is that I am not completely certain that any clear evidence is presented that the conformational switch is responsible for the loss of the phosphatase activity. There is a good correlation but the two things could be independent from one another.

My specific comments are listed below:

L69: you suggest that the mechanism is universal? Does this mean that all HK are controlled by pH? Other examples?

L102: gradually emerged. You did not show any "gradual" change at neutral pH. Is there a time course experiment?

L119: the title is too affirmative. From the data, it can only be said that H260 is involved (not "responsible").

L126: although diminished, the phosphatase activity is still present in the histidine mutant protein (see Fig. S3). Therefore, the histidine residue H260 is not absolutely required for this activity, an additional reason to tone down the title of this section.

L140: at or below pH 6.0 (not 5.0)

L155 and following: that the complex adopts distinct conformations at low and high pH is concluded from the comparison between two crystallographic studies performed independently, under different conditions, from different complexes (with and without BeF₃). Moreover, we compare here pH 5 and 5.6, a pH range in which the conformational change is not obvious from the NMR study (see Fig. 1d). It would have been preferable to run a "control" (i.e. neutral pH) in parallel with the experiment presented in this paper.

L182-200: the contribution of this part to the proposed model is not clear to me. In particular, it looks very speculative from L192.

L223: should it be "the catalytically inactive H243A EnvZ mutant does *not* generate phosphor-OmpR" ?

L273-279: this part should be moved to the Results section.

Minor comments :

L25-26: ...two-component system*s* that mediate** the cellular response...

L205: you mean "systemic" ?

L236: ...are enhanced in comparison to those in the WT EnvZ...

L313 : protein complexe*s*

L357: acrylami*d*e

L362: reference for LT2 strain?

L394: m*i*croscopy

In supplementary methods:

L11: ...containing plate to *select* the second DNA cross-change.

Figure 3b: coordinates of H243 and T247 should be included

Reviewer 1:

1. *I am only loosely acquainted with the TCS field and, as a more general interest reader, I found the introduction provided was rather scant in explanation. For example, I was expecting a compelling argument that a pH switch in phosphatase activity was established to be fundamentally important to the underlying biology. Also, I would suggest including the residue numbers of the relevant residues in this particular complex when introducing the field more generally. It makes a useful reference point for the less familiar reader.*

We appreciate the suggestion and have included comments about the effect of pH changes (e.g., acidification of the phagocytic vacuole) on the bacterial pathogenicity. We have also included the relevant residue numbers whenever appropriate to improve the readability of the manuscript.

2. *The authors do a good job of using NMR data to identify a change in conformation. I think the assignment of ^{19}F resonances in the RR468 complex is justified given the H3(N)F data. I'm surprised though when discussing the SIIS values that not all of the H-bond partners are included – what about M55NH and A84NH? The reason to be more inclusive is that changes in these H-bonds will also lead to chemical shift changes between complexes. Also the ^{19}F NMR spectra should be referenced to CFCl_3 , see https://en.wikipedia.org/wiki/Fluorine-19_nuclear_magnetic_resonance.*

We appreciate the comments by the reviewer and have included the effects of the hydrogen bonds involving the amide groups of M55 and A84 during the discussion of the SIIS values. Additionally, the chemical shifts of ^{19}F NMR signals are now referenced to CFCl_3 as suggested in the revised manuscript.

3. *I do not understand to use of the word “gradually” on line 102. Is formation of the complex slow, kinetically? I suspect not given the linewidth argument on lines 132-133, where incidentally the linewidths should be reported as quantitatively as achievable, because the differences are quite hard to see in the figure.*

The word “gradually” describes the intensity increase of new signals during the NMR titration and does not have any implication about binding kinetics. In order to avoid potential confusions, we have removed this word in the revised manuscript. Additionally, the pertinent signal linewidths are now specifically stated in the revised manuscript.

4. *The numbering of the fluorines in the pdb file provided and in the manuscript differ. I request that in both the authors label the fluorine bound to Mg as F_1 , the one H-bonded to K105 and A84 as F_2 , and the one H-bonded to M55 and T83 as F_3 . This is a new IUPAC convention that is currently in press and, as yet, unavailable to the authors. However, these changes would make the labeling consistent with the convention for the field in the future.*

We have updated the notation of ^{19}F NMR signals as requested.

5. *Line 110. To be consistent with Fig 1c “6.8” should read “7.0” since the top spectrum of Fig 1d and spectrum B of Fig 1c look indistinguishable.*

The pertinent text has been updated as requested.

6. *Line 117. Refer to Fig 1d here also.*

Reference to Fig 1b-d has been added.

7. *Line 148. The scenario regarding AMPPNP and ADP needs introducing more fully. Also, why was ADP not simply used for crystallization?*

Yes, ADP can be used for crystallization. However, in order to achieve unbiased structural comparison with the previous study by Casino et al (*Cell* 2009; 139: 325-336), we strived to maintain similar crystallization conditions except for pH. Therefore, AMPPNP was used instead of ADP for co-crystallization. This is now clearly stated in the revised manuscript.

8. *Line 149. Should read “close to symmetric”*

The text has been updated as requested.

9. *Line 155. This argument is central to the manuscript and needs better support with a figure than currently. I suggest either an overlay or side-by-side figure with 3dge.pdb highlighting the residues relevant to the model for how phosphoryl transfer is occurring. This will show the movement of H260 better than Fig 2c. I would also like the authors to use terminology other than sidechain flipping for the changes in H260 conformation. This terminology is easily misinterpreted as imidazole ring flipping rather than a larger scale movement of the entire sidechain.*

We appreciate the suggestion. Fig. 2c was indeed an overlay with 3dge.pdb. In order to avoid the confusion of describing the sidechain movement of H260 as ring “flipping”, we now describe this movement as “rotameric switching”. In Figure 2c, the rotameric χ_1 states of H260 in both structures are now explicitly labeled.

10. *Line 182. The comparison of chemical shift changes for the T264A mutant are interesting but the interpretation of their source is not a strong argument. I would suggest either cutting back this section to simply the behavior of the NMR, or leave it to a manuscript supported by a crystal structure of this mutant. It doesn't add greatly to the current manuscript, and the space gained by cutting it would enable a fuller introduction to the study. I would also drop Figure 4 and related text, as the data are not sufficiently strong to elevate this model over others.*

We appreciate the comment of the reviewer. In order to gain further evidence for our proposed model, we have now determined the crystal structure of the HK863-RR468 complex using the HK853 T264A mutant. The mutant complex structure reveals an inactive conformation as predicted from our NMR analysis, with H260 pointing away from the active site. The mutant complex structure overlays well with the wild-type HK853-RR468 complex at pH 5.0, with a backbone RMSD of 0.32

Å, but it differs from the active conformation of the HK853-RR468 complex (3DGE). Such a result, which provides strong support for our proposed catalytic model based on ^{19}F NMR studies, is described in the revised manuscript.

11. Line 202. Much of this section should be in the introduction. Indeed, maybe the Salmonella results should come before the Thermotoga results as the former set the scene for the relevance of the latter. I take it the Salmonella system does not crystallize well?

We appreciate the suggestion by the reviewer. The reviewer is correct in that the *Salmonella* system did not crystallize despite our best efforts. Therefore, we feel that a careful analysis of the structural and mechanistic insights from the *Thermotoga* system should precede the discussion of the functional consequence of the *Salmonella* system.

12. On a more general level, any reaction with an imidazole attacking a phosphate is likely to be pH dependent close to and below the pKa of the imidazole. A conformational switch is not needed, even though protonation of the imidazole may induce it. I think the discussion should take this into account unless clear evidence can be presented that the pH conformational shift is the result of protonation away from the active site. Also, the H-bonding between H260 and glycerol in the crystal should be discussed.

It is correct that a protonated imidazole ring will reduce the activity of H260 as the catalytic general base in the dephosphorylation reaction. However, in the absence of a conformational switch, a pure protonation change of the catalytic histidine residue invariably occurs on the fast time scale of NMR measurements (e.g., Sehgal AA, et al. Chem. Eur. J. 20, 6332-6338 (2014)). In contrast, our ^{19}F NMR study shows that the exchange rate between the active and inactive states of the HK853-RR468 complex occurs on the slow time scale. Such an observation argues that a simple change of the protonation state of H260 is insufficient to account for the ^{19}F NMR result, but it instead supports a model of pH-gated conformational switch on the μs -to- ms time scale. This is discussed in the revised manuscript.

We have also added comments about the hydrogen bond between glycerol and one of the two H260 residues in the methods section as requested. As both H260 residues in the HK853 dimer adopt the identical rotameric conformation, whereas only one of the two forms a hydrogen bond with the glycerol molecule (cryo-protectant), we are confident that our observed structural change of the H260 sidechain represents a *bona fide* conformational switch.

Reviewer 2:

An important body of research performed on different members of the HisKA family indicates that the conserved histidine residue acts as a general base to activate the nucleophilic water molecule (Casino et al, Cell, 2009, Huynh & Stewart, Mol Microbiol, 2011). Therefore it should be expected that the pH (7 vs 5, imidazole pKa 6.02) affects the phosphatase activity. However, the authors never state explicitly if H260 is the protonated residue (only in Figure 4 and its legend the proton is mentioned). To my knowledge, this paper presents the first report on the in vitro influence of the acidic pH on the HK domain, but this can be deduced from the literature, decreasing the novelty required for this journal.

As stated in our response to reviewer 1, it is correct that a protonated imidazole ring will reduce the activity of H260 serving as the catalytic general base in the dephosphorylation reaction; however, a *pure* protonation change of the catalytic histidine residue occurs on the fast time scale of NMR measurements, which is inconsistent with our ¹⁹F NMR result. Our ¹⁹F NMR study shows that the change of the fluorine signals occurs on the slow time-scale (μs-to-ms) during the pH titration; arguing against a *pure* protonation change of H260, but instead providing a strong support for a conformational switch that is dependent on the protonation state of H260. The notion of a pH-gated conformational switch is unequivocally supported by our crystallographic study at low pH, which has revealed a *catalytically-inactive* state with a large scale conformational change beyond H260 in comparison with the conformation of the *catalytically-active* state reported previously. Furthermore, the same *inactive* conformation is also captured in the T264A mutant, which provides the first direct structural evidence for the role of this catalytically important residue in the dephosphorylation reaction. None of these results can be deduced from the knowledge of H260 as the catalytic general base as the reviewer suggested. Hence, our observations reported in this manuscript are novel and provide significant insights into the regulation of the phosphatase activity of bifunctional histidine kinases.

The scope of the discovery is overestimated by the authors, as it is probably applicable only to some groups of histidine kinases. Moreover, both HKs studied in this work (HK853 and EnvZ) are from the same family (HisKA). For example, analysis of different mutants suggests that the phospho-accepting His residue is irrelevant for the phosphatase activity of HisKA_3 family, and given that the authors propose a central role for this histidine residue in the pH regulation, it might be possible that their findings are not reproduced among this entire superfamily.

In response to the reviewer's comments, we have specifically restricted our discussions to the HisKA family of the sensor histidine kinases in the revised manuscript. However, given that HisKA family represents ~80% of all sensor histidine kinases, we believe that our observations bear significant impact to the general field of two-component systems.

Besides, a pH sensor residing in a so conserved motif is difficult to justified, since

in such case all the HisKAs protein of an organism should be controlled by pH.

Just as changes of the redox potential have significant consequences on redox-sensitive enzymatic activities and biology in general, so would changes of pH affect pH-sensitive enzymes. As our studies have clearly demonstrated the altered phosphatase activity of HK853 and EnvZ, two prototypical members of the HisKA family of sensor kinases, in response to pH changes, we anticipate other members of the HisKA family of sensor kinases will display similar properties.

The experiments, especially the in vitro ones are very convincing. However a control of structure conservation at pH 5 is strictly needed, to rule out the possibility of enzymatic activity loss due to partial of full denaturation. Circular dichroism experiments should be enough. The in vivo evidence though, is scarce: only one qRT-PCR experiment, with only one mutant.

In response to the reviewer's comments, we have conducted circular dichroism experiments for HK853cp and its complex with RR468 at pH 8.0 and 5.0. The curves collected at pH 5.0 are identical to those collected at pH 8.0, supporting the structure integrity of the samples at different pHs. The circular dichroism results are now included as Fig. S3 in the *Supplementary Materials*.

The finding is highly important to scientist in the field who should consider pH as an important variable to take into account for histidine kinase enzymatic activities, but probably not of broad interest.

We respectfully disagree with the reviewer about this. Understanding how the bacterial two-component system responds to different environmental cues, including pH changes, such as those encountered during the phagocytosis of the *Salmonella* infection, is fundamental to our understanding of bacterial pathogenesis and bacterial adaptation to the host environment. Insights gained from such studies undoubtedly would be of broad interest to researchers in the field as well as general readers of *Nature Communications*.

In both in vivo experiments, the qRT-PCR and the infection a mutant with which does not bind ATP and therefore with no kinase activity but with intact phosphatase activity should be included.

We appreciate the desire to create a phosphatase-only EnvZ mutant, but such an enzyme will have no substrate (i.e., phospho-OmpR) to catalyze, and the outcome will not be different from the H243A EnvZ mutant that we use as the background control.

In the qRT-PCR experiment a better way to show the data would be to normalize the result to the wild-type strain in neutral pH.

Although it is conceptually possible to normalize everything against the wild-type strain in neutral pH, in practice, the large variation in cell conditions, growth rates, and background signals of conditions between pH 7 and pH 5 renders such comparison unreliable. Instead, by normalizing the results of different strains under each pH condition individually against the background signal in the H243A EnvZ

mutant with no kinase activity, we have been able to obtain consistent and statistically meaningful results presented in this study.

An experiment analyzing the in vivo phosphorylated and unphosphorylated forms of OmpR in cultures grown under different pH conditions in wild-type and mutants Salmonella strain could increase the in vivo evidence.

This is a good suggestion, but it would require specific antibodies against phospho-OmpR and OmpR. To the best of our knowledge, only one company (Antibody Research Corporation) claimed to supply the OmpR antibody but not the P~OmpR antibody. However, we have not received any response from the company after repeated requests. The current qRT-PCR experiment was designed according to the protocol described in the study of *PLoS Biol.* 13, e1002116 (2015).

Apart from NMR and crystallographic experiments that are usually robust enough and accepted without repetitions, the authors perform qRT-PCR in which they do the experiment twice. They also perform an experiment using confocal microscopy, which they performed twice. The description of the quantification is not in the Materials and Method Section, only in the Figure Legend. No statistical analysis is applied at all in this report. However the differences shown seems to be big enough.

We apologize for the typo here. The qRT-PCR experiments were actually performed four times independently (in two different batches), and the mean values and standard deviations (SD) of these experiments were calculated from the data of four repeats. This information is updated in the revised manuscript.

Despite the report is in general well-written the Discussion Section is very poor and should be enriched, putting the new results in context with other report of the HK853-RR468 and EnvZ-OmpR and other HK-RR systems.

The discussion section has now been expanded to compare our result with published structural analyses of HK-RR systems.

An important point that has not been addressed in the manuscript is the influence of pH on the autophosphorylation and phosphotransfer activities of HK853 and EnvZ. If pH mediated a switch, it would be expected to regulate the autophosphorylation and phosphatase activities in opposite directions (i.e. at pH 5.0 HK853 should be an active kinase).

As the current study focuses on the regulation of the phosphatase activity of HK853 and EnvZ, the investigation of the kinase activity, although important, is beyond the scope of this manuscript.

Regarding the crystal structure presented, there are not enough details provided on the interactions that might stabilize the position of H260 at pH 5.0. Also, based on this structure, the authors propose that the configuration adopted by H260 would prevent it from participating in the phosphatase activity, but they do not discuss the possibility that, in fact, protonation itself precludes the catalytic role of H260 and that

its side chain flipping is just aleatory or a consequence of relocating the new positive charge introduced in the imidazole ring.

H260 in the altered rotameric conformation at pH 5.0 is likely stabilized by water-mediated hydrogen bonds with neighboring residues, which are not visible at the current resolution limit. We agree with the reviewer that in general we cannot separate the change of the protonation state of H260 from its conformational switch as observed in the crystal structure, but our NMR studies clearly show that a simple change of the histidine protonation state, which occurs in the fast-exchange limit of NMR measurements, is inconsistent with the slow-exchange behavior of the fluorine signals at different pH values. These observations highlight the importance of a conformational switch for regulating the phosphatase activity of HK853.

The conclusion of the role of the H260 side chain flipping comes from the comparison of the obtained structure at pH 5.0 with the one from Casino et al, Cell, 2009 at pH 5.6. To correlate structure and phosphatase activity the authors therefore should also perform the activity assay at pH 5.6.

The population of a specific conformation captured in the crystal structure can differ significantly from that in the solution state, as the crystallization process often selectively enriches the conformation most prone to crystallization. For example, in our previous study of LpxC-inhibitor complexes (*Nat Commun.* 2016; 7: 10638), the crystal structures captured a single conformation of the inhibitor in the *Aquifex aeolicus* LpxC/LPC-011 complex and in the *Aquifex aeolicus* LpxC/LPC-023 complex. However, solution NMR studies readily revealed alternative inhibitor conformations in solution (with the alternative population of up to 25%). Likewise, our ¹⁹F NMR result shows that the HK853-RR468 complex exists as a mixture of two conformations with similar populations at pH 5.6 in solution. Therefore, such a pH condition (pH=5.6) is not the optimal choice for detailed activity analysis of individual conformational states in solution.

Finally, since the authors do not address the possibility of an enhanced kinase/phosphotransferase activity at acidic pH, they do not consider that the conformation crystallized might represent a snapshot of an intermediate state in the phosphotransfer reaction. In a paper published by Podgornaia et. al., Structure, 2013 the structure of a complex between rewired HK853 and RR468 is presented, and the H260 adopts a rotamer that points away from the active site and the authors propose that the complex might represent an earlier state of the phosphatase reaction (when the phosphate is still bound to the RR) or the end of the phosphotransfer reaction (just prior to the complex dissociation). So, in this manuscript the authors should make a reference to the mentioned article and evaluate the similarities/differences between the complexes.

We appreciate the suggestion and have now commented on the work by Podgornaia and co-workers in the Discussion section as the following:

Interestingly, although the catalytically inactive conformation reported here has not

been previously captured in wild-type enzymes, it has been observed in a “rewired” HK853^{cp*}-RR468* complex, in which the interface residues of HK853 and RR468 from *Thermotoga maritima* were mutated to match that of the RhoR-RhoB two-component system in *E. coli*, respectively ²⁵. As these mutations also perturbed the arrangement and relative orientation of the histidine kinase with respect to the response regulator, the critical factor that trapped the chimeric complex into the inactive state has not been isolated. In comparison, by combining solution NMR and crystallography studies of the wild-type HK853-RR468 complex, we have been able to establish the critical role of pH in regulating the conformation-dependent phosphatase activity of the HisKA family of bifunctional histidine kinases. Whether the inactive conformation of the wild-type HK853-RR468 complex observed at low pH could resemble snapshots of transient intermediates of the phosphotransfer reaction as previously suggested for the rewired complex ²⁵ remains to be investigated.

- Page 9, line 183-184. “It has been reported that the T264A mutant of HK853 abolishes the phosphatase activity, though the molecular mechanism remains unclear”. Actually, there is evidence that support that the conserved Thr or Asn residue in the E/DxxN/T motif coordinates the nucleophilic water for phosphatase activity (for references see Huynh, Stewart, 2011). The authors then propose the participation of a water molecule, so references to previous observations should be made.

We appreciate the suggestion and have included this reference in the revised manuscript.

- Page 9, lines 190-192. “... the critical chemical entity participating in the K105RR468-fluorine (F2) interaction was absent in the T264A mutant of the HK853-RR468 complex”. Is not it possible to evaluate the presence of the salt bridge by NMR, as it was done with the free RR?

The chemical entity suggested here is the catalytic water molecule that forms hydrogen bonds with T264 and the catalytic general base H260, with the sidechain of H260 adopting the *trans* χ^1 rotamer in an catalytically competent conformation. The experimental detection of the direct interaction between such a water molecule and the fluorine atom is difficult by NMR, but the preservation and perturbation of such an interaction can be readily detected by the readout of the fluorine chemical shift.

The loss of a catalytic water molecule is further supported by the new crystal structure of the T264A mutant of the HK853 -RR468 complex. In this structure, the sidechain of H260 in HK853 flips away from the active site in a *gauche*- χ^1 rotameric state and is no longer able to coordinate the catalytic water molecule. This, together with the loss of the hydrogen bond from T264, will likely result in the loss of the catalytic water molecule as proposed previously, generating a fluorine signal of the F₂ atom of BeF₃⁻-RR468 that corresponds to the *apo* state.

- It should be informed if the initial adhesion of the different *Salmonella* strains is the

same, in order to prove that the results of the macrophage infectivity assay at 4 h is a consequence of different ability to survive within the acidic compartment and not in differences in the initial adhesion and ability to invade the cell.

We acknowledge that we cannot differentiate the difference between initial adhesion and the ability of the bacteria to invade the cells. Hence, the word “invasion” was replaced with “infectivity” in the revised manuscript.

- The figure legends (the main and the supplementary ones) should be self-explicative. All the important information about the experiment should be summarized here. Please check the Figure Legends Section of the checklist.

The figure legends have been updated according to the request.

- Page 12, lines 250-259. This information is not relevant in this context.

The text mentioned here summarizes the current knowledge about the regulatory mechanisms of the two component systems. Therefore, we feel that it is appropriate to have this discussion in order to put our discovery in the context of other characterized regulatory mechanisms of two component systems.

- Figure S3. The phosphatase experiment using the WT HK853 does not show any RR-P at t=0. It would be better if the authors provided a figure where it is shown that the reaction was performed with the RR phosphorylated at the beginning of the assay.

The zero time data (t=0) were shown at the second lane of the image. Due the spontaneous hydrolysis process, it is not feasible to generate a sample of clean phospho-RR468 without the appearance of apo-RR468.

- Page 13, line 263-279. It is not clear either if the modulation of the infectivity by the EnvZ/OmpR pathway is a new finding (which by the way should be mentioned in the Result not the Discussion section) or if it is something already known.

It is known that the EnvZ/OmpR two-component system controls the gene expression of the *Salmonella* Pathogenicity Island (SPI) 2 and plays an important role in bacterial pathogenicity. Recently, Chakraborty and co-workers demonstrated that the cytoplasmic pH of *Salmonella* is reduced during phagocytosis, and blocking acidification results in defective secretion of SPI-2 proteins, though the molecular basis of how the EnvZ/OmpR system responds to the change of pH has remained unknown. Our study thus provides the first molecular insight into how EnvZ/OmpR controls the downstream events by modulating the EnvZ phosphatase activity in response to pH changes.

The pertinent text has now been moved to the Results section as suggested.

- Page 5, line 103. The fact that the phosphatase activity resides in the DHp subdomain should be included as a references (see several report of the review Huynh & Stewart, Mol Microbiol, 2011).

We appreciate the reviewer’s suggestion and have added the relevant references in the introduction part.

- Page 9, line 179. "... suggest that the bifunctional activity of HK853 is regulated by a pH-gated ...". Strictly, only the phosphatase activity was evaluated, not the kinase.

We have revised the "bifunctional" as "phosphatase" here.

- Page 10, line 205. "systemic" instead of "systematic".

We have revised the word as "systemic".

- Page 11, line 223. "... the catalytically inactive H243A EnvZ mutant does generate phospho-OmpR". This sentence does not make any sense, given that the mutation H243 does not allow the autophosphorylation to occur, and therefore there is no generation of OmpR-P. In fact, the authors also wrote "... ablation of the kinase activity in the H243A EnvZ mutant eliminated the production of phospho-OmpR" (page 13, lines 274-275). Probably the authors wanted to say "does not generate", instead.

We apologize for this typo. The revised sentence now reads as "the catalytically inactive H243A EnvZ mutant does not generate phospho-OmpR".

- Page 11, lines 235-236. The authors state that the T247 mutant is deficient in the phosphatase activity. They do not demonstrate it in vitro but there are reports of this statement, so they should provide the corresponding references.

We have added the relevant references in the introduction part as requested (Huynh, et al. Mol Microbiol, 2011, Willett JW, et al. Plos Genet, 2012; Huynh, et al. PNAS, 2010, Casino, et al. NC, 2014).

- Page 16, line 334. A citation of the report where this crystallization condition was found should be included.

The reference to the original crystallization condition of the wild-type HK853-RR468 complex is now included in the *Method* section.

- Figure 3a. EnvZ H243A and EnvZ T247A element are not clear. As it is drawn it seems inhibition.

Yes. The H243A and T247A mutants abolished the kinase and phosphatase activities of EnvZ respectively. This is now explicitly stated in the figure legend.

Reviewer 4

Although the work is interesting and well presented, and the data are convincing regarding the pH-mediated phosphatase inhibition and the pH-induced conformational switch, my main concern is that I am not completely certain that any clear evidence is presented that the conformational switch is responsible for the loss of the phosphatase activity. There is a good correlation but the two things could be independent from one another.

The contribution of the pH-induced conformational switch to the phosphatase activity is based on two key observations: the crystal structure of a catalytic inactive conformation at low pH and the slow exchange rate between the catalytically inactive and active states observed by fluorine NMR. Neither of these observations can be explained by a simple pH-mediated protonation of the catalytic general base. Therefore, we feel that the pH-induced conformational switch is the most suitable explanation here.

L69: you suggest that the mechanism is universal? Does this mean that all HK are controlled by pH? Other examples?

As stated in response to previous reviewers, we have narrowed the claim to the HisKA family in the revised manuscript. The conservation of the HE/DxxT/N motif in this family suggests that these sensor HKs should also respond to pH changes.

L102: gradually emerged. You did not show any “gradual” change at neutral pH. Is there a time course experiment?

The word “gradually” has been removed to avoid confusions.

L119: the title is too affirmative. From the data, it can only be said that H260 is involved (not “responsible”).

We have made the change as requested.

L126: although diminished, the phosphatase activity is still present in the histidine mutant protein (see Fig. S3). Therefore, the histidine residue H260 is not absolutely required for this activity, an additional reason to tone down the title of this section.

The section title has been changed as requested.

L140: at or below pH 6.0 (not 5.0)

Our NMR data show that the populations of the two conformational states are about equal at pH 5.8; the population of the catalytically inactive state becomes the predominant state at pH 5.0, with an estimated population of ~80-90%.

L155 and following: that the complex adopts distinct conformations at low and high pH is concluded from the comparison between two crystallographic studies performed independently, under different conditions, from different complexes (with and without BeF₃). Moreover, we compare here pH 5 and 5.6, a pH range in which the

conformational change is not obvious from the NMR study (see Fig. 1d). It would have been preferable to run a "control" (i.e. neutral pH) in parallel with the experiment presented in this paper.

We agree with the reviewer about the limitation of the protein crystallization conditions. Unfortunately, protein crystallization remains to be an art with many factors beyond our control, and despite numerous attempts, we have not been able to obtain the complex crystals at neutral pH. This is why solution NMR information is essential in our investigation as it *bridges* the gap unfilled by the crystallographic studies.

L182-200: the contribution of this part to the proposed model is not clear to me. In particular, it looks very speculative from L192.

In response to the reviewer's comments, we have now obtained the crystal structure of the T264A HK853-RR468 complex. This structure offers an excellent explanation of the observed fluorine NMR signals and provides strong experimental support for our proposed model summarized in the revised Figure 3.

*L223: should it be "the catalytically inactive H243A EnvZ mutant does *not* generate phosphor-OmpR" ?*

We apologize for this typo. This has been corrected in the revised manuscript.

L273-279: this part should be moved to the Results section.

The pertinent text has been moved to the Results section as suggested.

*L25-26: ...two-component system*s* that mediate** the cellular response...*

The text has been updated as suggested.

L205: you mean "systemic"?

Yes. The text has been updated accordingly.

L236: ...are enhanced in comparison to those in the WT EnvZ.

The text has been updated as suggested.

*L313 : protein complexe*s**

The text has been updated as suggested.

*L357: acrylami*d*e*

The typo has been fixed in the revised manuscript.

L362: reference for LT2 strain?

The reference has been added (McClelland M, et al. *Nature*, **2001**; Vikram A, et al. *Int. J. Food. Microbiol.* **2011**)

*L394: m*i*croscopy*

The typo has been fixed.

*In supplementary methods: L11: ...containing plate to *select* the second DNA cross-change.*

The text has been updated as suggested.

Figure 3b: coordinates of H243 and T247 should be included

The figure (revised Figure 4b) has been updated as requested.

REVIEWERS' COMMENTS:

Reviewer #2 (Remarks to the Author):

The authors have incorporated two new experiments: (1) the crystal structure of the HK853cp_T264A – BeF₃⁻ - RR468 complex, in order to gain structural evidence for the mechanism proposed of phosphatase regulation by pH and (2) a control with circular dichroism spectra of HK853cp and in complex with RR468 at pH 8 and 5, to rule out the loss of phosphatase activity at pH5 due to partial or full denaturation. While the circular dichroism experiments demonstrate that the overall structure is not perturbed by pH, the analysis of the new crystal structure is scarce. A well justified hypothesis about what should be observed in HK853cp_T264A – BeF₃⁻ - RR468 complex is needed.

All the reviewers agreed in that the authors clearly demonstrated that the pH affects the phosphatase activity, but have our concern about the evidence that the bifunctional histidine kinase is a pH sensor, especially taking into account that the conserved histidine residue, which is essential in this process changes its protonation state between pH 7 and 5. In response to this issue, the authors have improved the explanation about which is the conformational change that supports the idea of a pH sensor. The conformational change consists in the sidechain flipping, in my opinion quite a local change to be consider as sensing. The notion of the pH sensor is supported by (1) the alternative position of the histidine sidechain in HK853cp – BeF₃⁻ - RR468 and HK853cp_T264A – BeF₃⁻ - RR468 complexes crystal structures and (2) the fact that the histidine protonation occurs on a fast timescale, while the sidechain flipping occurs in a much slower timescale (μ s-to-ms). The other line of evidence that could reinforce the hypothesis of the pH sensor is the biological one. I have mentioned some concerns about the biological experiments, none of which have been satisfactorily addressed. According to the exposed argument I consider that the evidence that support that the bifunctional histidine kinase belonging to the HisKA family are pH sensor is not strong and broad enough for this journal.

Additional issues:

Line 74. "consisting of" instead of "including"

The sentence between lines 75 and 78 is confusing.

Responses to REVIEWERS' COMMENTS:

Reviewer #2 (Remarks to the Author):

The authors have incorporated two new experiments: (1) the crystal structure of the HK853^{cp}_T264A – BeF₃⁻ - RR468 complex, in order to gain structural evidence for the mechanism proposed of phosphatase regulation by pH and (2) a control with circular dichroism spectra of HK853^{cp} and in complex with RR468 at pH 8 and 5, to rule out the loss of phosphatase activity at pH5 due to partial or full denaturation. While the circular dichroism experiments demonstrate that the overall structure is not perturbed by pH, the analysis of the new crystal structure is scarce. A well justified hypothesis about what should be observed in HK853^{cp}_T264A – BeF₃⁻ - RR468 complex is needed.

The purpose of the T264A HK853^{cp}-BeF₃⁻-RR468 complex structure is to support the proposed catalytic mechanism shown in Fig. 3c. As the F₂ fluorine signal of the T264A HK853^{DHP}-BeF₃⁻-RR468 complex (Fig. 1c, spectrum E) shows a chemical shift identical to the *apo* state (substrate state) of BeF₃⁻-RR468 complex, we anticipate that any direct or indirect interactions between the F₂ fluorine and HK853 will be abolished in the T264A HK853 mutant. Our crystal structure of the T264A HK853^{cp}-BeF₃⁻-RR468 complex structure supports this hypothesis by showing the total loss of water-mediated interactions between T264 and H260 with the F₂ fluorine atom due to (1) the T264A mutation that removes the hydroxyl group and (2) the *gauche*- χ 1 rotameric configuration of H260 that prevents its engagement of the water mediated interaction with the F₂ fluorine atom. These two changes would return the chemical environment of the F₂ fluorine atom to a state similar to the *apo* BeF₃⁻-RR468, despite the formation of the T264A HK853^{cp}-BeF₃⁻-RR468 complex. In order to explain this connection more explicitly, we have revised Figure 3 to illustrate these concepts.

All the reviewers agreed in that the authors clearly demonstrated that the pH affects the phosphatase activity, but have our concern about the evidence that the bifunctional histidine kinase is a pH sensor, especially taking into account that the conserved histidine residue, which is essential in this process changes its protonation state between pH 7 and 5. In response to this issue, the authors have improved the explanation about which is the conformational change that supports the idea of a pH sensor. The conformational change consists in the sidechain flipping, in my opinion quite a local change to be consider as sensing. The notion of the pH sensor is supported by (1) the alternative position of the histidine sidechain in HK853^{cp} – BeF₃⁻ - RR468 and HK853^{cp}_T264A – BeF₃⁻ - RR468 complexes crystal structures and (2) the fact that the histidine protonation occurs on a fast timescale, while the sidechain flipping occurs in a much slower timescale (μ s-to-ms). The other line of evidence that could reinforce the hypothesis of the pH sensor is the biological one. I have mentioned some concerns about the biological experiments, none of which have been satisfactorily addressed.

According to the exposed argument I consider that the evidence that support that the bifunctional histidine kinase belonging to the HisKA family are pH sensor is not strong and broad enough for this journal.

The notion of a pH-dependent conformational switch is well supported by (1) the observed conformational differences in our crystal structures (the WT HK853cp-BeF₃⁻-RR468 complex and the T264A HK853cp-BeF₃⁻-RR468 complex) determined at low pH and the previously reported crystal structure, and (2) the slow timescale (*μs-to-ms*) of the conformational switch. Additionally, the observed conformational switch is not limited to local variations and is in fact accompanied by large scale domain movements as illustrated in Figure 2c and 2d. None of these observations is consistent with a simple change of the H260 protonation state, which does not involve any conformational changes and occurs on the fast timescale of NMR measurements.

We are unable to carry out the suggested biological experiment to determine the phospho-OmpR level in cells due to the lack of commercial antibodies against phospho-OmpR.

Additional issues:

Line 74. “consisting of” instead of “including”

The “including” is changed to “involving”. As the conformational change is not restricted to H260, we feel the word “involving” is the better description here.

The sentence between lines 75 and 78 is confusing.

We have revised the relevant sentence to improve the clarity.